# A system wide approach to managing zoo collections for visitor attendance and in situ conservation

Andrew Mooney [1,2], Dalia A. Conde [2,3], Kevin Healy[4,5] & Yvonne M. Buckley [1*]

Zoos contribute substantial resources to in situ conservation projects in natural habitats using revenue from visitor attendance, as well as other sources. We use a global dataset of over 450 zoos to develop a model of how zoo composition and socio-economic factors directly and indirectly influence visitor attendance and in situ project activity. We find that zoos with many animals, large animals, high species richness (particularly of mammals), and which are dissimilar to other zoos achieve higher numbers of visitors and contribute to more in situ conservation projects. However, the model strongly supports a trade-off between number of animals and body mass indicating that alternative composition strategies, such as having many small animals, may also be effective. The evidence-base presented here can be used to help guide collection planning processes and increase the in situ contributions from zoos, helping to reduce global biodiversity loss.

[1] School of Natural Sciences, Zoology, Trinity College Dublin, Dublin, Ireland. [2] Conservation Science Alliance, Species360, 7900 International Drive, Suite 1040, Bloomington, MN 55425, USA. [3] Center on Population Dynamics (CPop), Department of Biology, University of Southern Denmark, Campusvej 55, 5230, Odense M, Denmark. [4] School of Biology, University of St Andrews, Scotland, UK. [5]Present address: Ryan Institute, School of Natural Sciences, National University of Ireland Galway, Galway, Ireland. *email: buckleyy@tcd.ie

Modern zoos contribute to the recovery and conservation of threatened species through ex situ breeding within institutions[1] and through substantial contributions to in situ conservation projects in natural habitats[2]. In order to fulfill their multiple roles, zoo collections must attract recreational visitors[3] and perceived visitor preferences have fueled the belief that large vertebrates, particularly mammals, are necessary in order to attract visitors[4]. However, compared to smaller species, large animals are often costlier to maintain, prove more difficult to breed in captivity, require larger enclosure sizes[5], and raise ethical and welfare issues[6]. As the global zoo community has a limited capacity[7], zoos have been encouraged through conservation objectives to shift their focus towards smaller-bodied species (particularly amphibians, invertebrates and fish), native species, threatened species and specialise on fewer species[8,9]. However, this compositional shift could result in reduced visitor attendance, lowering the economic return and consequently in situ conservation investment[10,11].

The global zoo and aquarium community fulfils several objectives, including conservation, education, research and entertainment[3]. These multiple roles can place competing demands on the composition of zoo collections as public preferences do not always align with conservation priorities. Collectively, the global zoo and aquarium community attracts >700 million visitors every year and invests >$350 million in wildlife conservation in situ, representing the third largest conservation organisation contributor globally[2]. These in situ conservation activities are primarily funded by paying visitors, in conjunction with other sources, and the popularity of institutional collections (in terms of the species within the collection) is positively correlated with attendance[12]. There is evidence for the flagship approach of using popular, large vertebrates in zoo collections to drive public education and in situ conservation fundraising[13], helping to protect other species and habitats[14,15]. However to our knowledge, work to date has yet to unequivocally link collection species composition to attendance worldwide, with most studies limited by the range of species, institutions and countries assessed[10,12]. Socio-economic variables also drive attendance[16] but the relative influence of socio-economic and collection composition variables on attendance have not been assessed. While the direct effects of various factors on attendance have been the focus of previous studies, such approaches fail to capture the complexity of potential indirect drivers of, and trade-offs for, visitor attendance. A framework linking the direct and indirect effects of collection composition variables on conservation outcomes, such as in situ contributions, would allow for more informed collection planning decisions and policy formation.

We test whether collection composition and socio-economic variables affect both institutional attendance (458 zoos worldwide) and in situ contributions (subset of 119 zoos). We use structural equation modelling (SEM) to test the determinants of both visitor attendance and in situ conservation contributions as part of a system of species and zoo characteristics and broader socio-economic variables (Table 1). We use vertebrate composition data from Species360 member institutions, in conjunction with the attendance information from the International Zoo Yearbook (IZY) and in situ project contribution reports from the Association of Zoos and Aquariums (AZA; Methods section). We find that zoos with many animals, large animals, high species richness (particularly of mammals) and which are dissimilar to other zoos achieve higher numbers of visitors and contribute to more in situ conservation projects. However, the model strongly supports a trade-off between number of animals and body mass, indicating that alternative composition strategies, such as having many small animals, may also be effective.

## Results

**Total effects of composition and socio-economic variables**. We found that zoos with high attendance contribute to more in situ conservation projects (Fig. 1). Zoo area and the proportion of threatened species are also positively correlated with in situ conservation projects, albeit these effects are weaker than attendance (Fig. 1). Collection composition variables (total no. of animals, total species richness, mammal species richness, compositional dissimilarity and species body mass) are more important in determining attendance than socio-economic variables (population density and gross domestic product [GDP]).

**Direct and indirect effects of variables**. The total effects of each variable (Fig. 1) are composed of direct and indirect effects (Fig. 2); for example the strong direct effect of body mass on attendance is weakened in the total effect of body mass on attendance due to negative effects of body mass on species richness, total number of animals and dissimilarity. Of the collection composition variables, the total number of animals had the largest direct positive effect on attendance, followed by the abundance

**Table 1 Description of the variables used within the structural equation models.**

| Variable | Description |
| --- | --- |
| Attendance | Annual institution attendance (2015) |
| Species richness | Total number of species per institution (2017) |
| Total animals | Total number of individual animals per institution (2017) |
| Mammal species richness | Total number of mammalian species per institution (2017) |
| Institution area | Institutional area in hectares (ha; 2015) |
| Threatened species proportion[a] | The proportion of the International Union for the Conservation of Nature (IUCN) Red List of Threatened Species™ 'threatened' species ('Critically Endangered', 'Endangered' and 'Vulnerable') per institution (2017) |
| Mean species body mass[a] | The mean species body mass per institution (g; 2017) |
| Diversity | Brillouin index measure of within collection diversity (alpha diversity; 2017) |
| Dissimilarity | The mean Raup–Crick dssimilarity index per institution, measuring compositional dissimilarity between collections (2017) |
| GDP | Gross domestic product (US$; 2015) |
| National population size | National population size for each country (2015) |
| 10 km Population | Estimated population count within a 10 km radius of the institution (2015) |
| In situ contributions | The annual number of field conservation programmes in which individual AZA member institutions contribute to in some capacity (2015) |

[a]Weighted for species abundance per institution

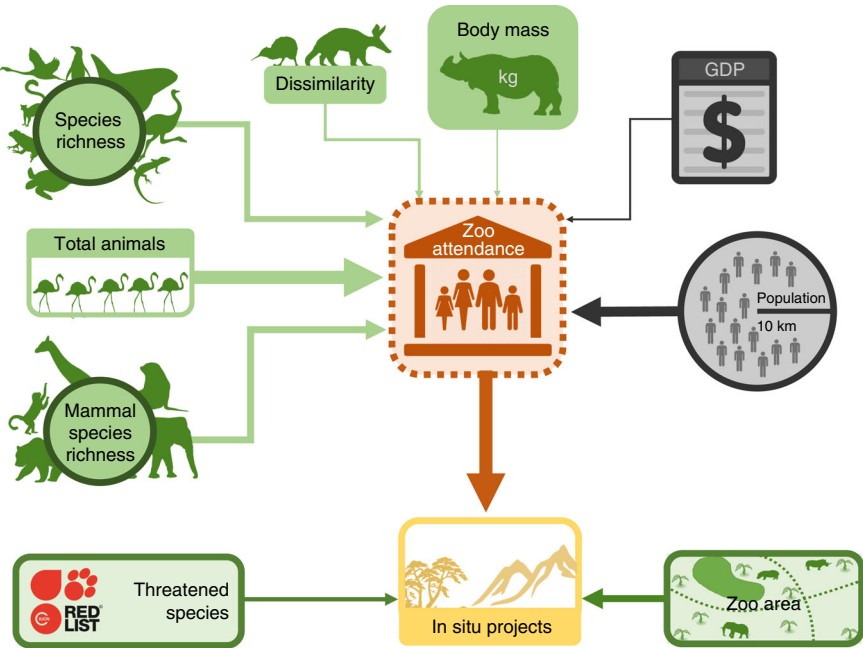

**Fig. 1 Total effects of institutional variables and socio-economic variables on visitor attendance and in situ contributions.** This simplified version of the SEM framework shows the total effects of explanatory variables on attendance and in situ contributions as arrows with line width representing the standardised relative effect sizes. All total effects were positive. Grey boxes represent socio-economic variables and green boxes represent institutional variables. Source data: total effect sizes were quantified using the Supplementary Code and Supplementary Data 1 and 2 provided.

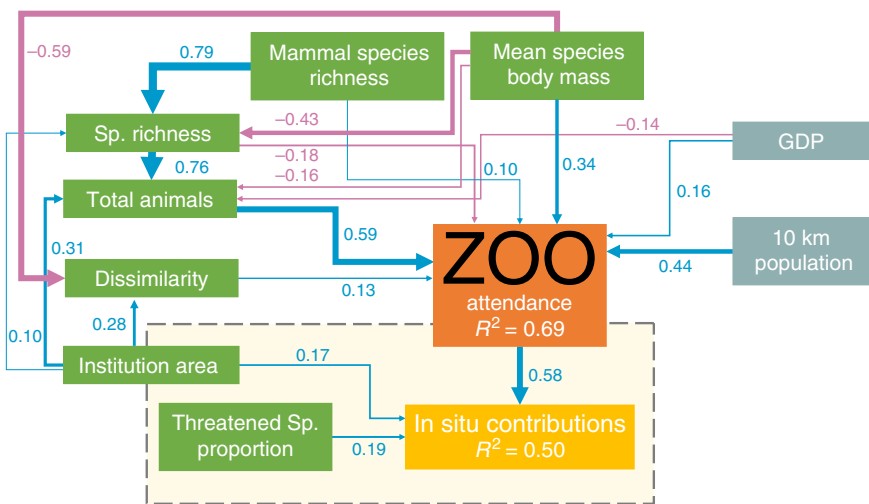

**Fig. 2 The SEM framework showing direct and indirect connections between institution attendance ($n = 458$), in situ contributions ($n = 119$), and various institutional and socio-economic variables.** Path coefficients shown are standardised. The yellow box indicates the additional pathways included for the 119 institutions for which in situ investment data was available. Blue arrows represent positive effects and pink arrows represent negative effects. Line width represents relative effect sizes. Grey boxes represent socio-economic variables and green boxes represent institutional variables. Source data: model structure and coefficients were determined using the Supplementary Code and based on Supplementary Data 1 and 2 provided.

weighted mean species body mass, with compositional dissimilarity and mammal species richness having smaller direct positive effects (Fig. 2).

Consistent with previous findings[16], we found that human population size and GDP had positive direct effects on institutional attendance, however, GDP also had a negative indirect effect on attendance via a negative effect on total number of animals (Fig. 2). Contrary to expectations[12], threatened species representation had no direct or indirect effects on attendance. Species richness had a strong positive indirect effect on attendance mediated by total number of animals, but species richness had a smaller direct negative effect on attendance. Mammal species richness alone had a direct positive effect on attendance, as well as multiple indirect positive effects through the total number of animals. Mammal species richness also had a small negative effect on attendance mediated by species richness. However, the total effect of mammal species richness on attendance was greater than that of the overall species richness (Fig. 1 and Table 2), suggesting mammals are more important in driving visitor attendance than other taxonomic groups.

**Table 2 Direct and total standardised effect sizes and proposed interpretations for both the attendance and in situ models.**

| | *p* Value | Direct effect (SE) | Total effect | Interpretation |
|---|---|---|---|---|
| Attendance model | | | | |
| Attendance ($R^2 = 0.689$) | | | | |
| Attendance~total animals | <0.001 | 0.587 (0.041) | 0.587 | Attendance is positively correlated with total number of animals in an institution |
| Attendance~10 km population | <0.001 | 0.444 (0.034) | 0.444 | Attendance is positively correlated with the local population size (10 km radius) surrounding an institution |
| Attendance~body mass | <0.001 | 0.340 (0.030) | 0.062 | Attendance is positively correlated with mean species body mass for an institution |
| Attendance~GDP | <0.001 | 0.163 (0.027) | 0.083 | Attendance is positively correlated with national GDP |
| Attendance~dissimilarity | <0.001 | 0.125 (0.031) | 0.125 | Attendance is positively correlated with collection dissimilarity |
| Attendance~mammal species richness | 0.021 | 0.102 (0.044) | 0.309 | Attendance has a small, but positive correlation with number of mammal species present in an institution |
| Attendance~species richness | 0.004 | −0.184 (0.064) | 0.262 | Attendance is directly negatively correlated with institutional species richness |
| Total animals ($R^2 = 0.783$) | | | | |
| Total animals~species richness | <0.001 | 0.759 (0.050) | 0.759 | The total number of animals in an institution is positively correlated with institutional species richness |
| Total animals~institution area | <0.001 | 0.309 (0.045) | 0.382 | The total number of animals in an institution is positively correlated with institutional area |
| Total animals~GDP | 0.047 | −0.136 (0.069) | −0.136 | The total number of animals in an institution is negatively correlated with national GDP |
| Total animals~body mass | <0.001 | −0.157 (0.036) | −0.483 | The total number of animals in an institution is negatively correlated with the mean species body mass of an institution |
| Species richness ($R^2 = 0.678$) | | | | |
| Sp. richness~mammal species richness | <0.001 | 0.790 (0.067) | 0.790 | Institutional species richness is strongly positively correlated with institutional mammal species richness |
| Sp. richness~institution area | 0.017 | 0.096 (0.040) | 0.096 | Institutional species richness is positively correlated with institutional area |
| Sp. richness~body mass | <0.001 | −0.429 (0.043) | −0.429 | Institutional species richness is negatively correlated with the mean species body mass of an institution |
| Dissimilarity ($R^2 = 0.257$) | | | | |
| Dissimilarity~institution area | <0.001 | 0.277 (0.051) | 0.277 | Collection composition dissimilarity is positively correlated with institutional area |
| Dissimilarity~body mass | <0.001 | −0.593 (0.077) | −0.593 | Collection composition dissimilarity is negatively correlated with the mean species body mass of an institution |
| In situ model | | | | |
| In situ contributions ($R^2 = 0.496$) | | | | |
| In situ~attendance | <0.001 | 0.583 (0.074) | 0.583 | Institutional in situ contributions are positively correlated with institutional attendance |
| In situ~threatened species proportion | 0.004 | 0.189 (0.066) | 0.189 | Institutional in situ contributions are positively correlated with the proportion of threatened species in an institution |
| In situ~institution area | 0.015 | 0.169 (0.069) | 0.320 | Institutional in situ contributions are positively correlated with institutional area |

Also provided are $R^2$ values, standard errors and *p* values
Relationships are ranked according to direct effect size magnitude. Model results presented reflect abundance adjusted models. Only the in situ component of the in situ model is reported as all other pathways were analogous to the attendance model. All estimated *p* values and quantities generated were derived using SEM, as outlined in the Supplementary Code and Supplementary Data 1 and 2 provided

## Discussion

No support was found for linking species body mass directly with in situ project activity. This suggests that in situ activity does not directly rely on the presence of large vertebrates, instead the effect of body mass is mediated by institutional attendance. We conclude that the absence of large vertebrates from collections may not necessarily result in reduced in situ project activity, presuming institutional attendance can be maintained in their absence through an increase in collection dissimilarity, species richness and/or total number of animals.

Additional compositional options may also be considered to increase the in situ contributions of institutions. The direct link between the proportion of threatened species present and institutional in situ contributions suggests that greater institutional investment in threatened species ex situ is positively correlated with higher in situ conservation activity. This may be through the integration of species-specific in situ and ex situ conservation

actions as encouraged in the IUCN Species Survival Commission (SSC) Guidelines on the Use of Ex situ Management for Species Conservation and the contemporary One Plan approach to species conservation suggested by the IUCN SSC Conservation Planning Specialist Group[17,18]. Interestingly, the proportion of threatened species was not an important factor in driving attendance, which may contradict evidence of perceived species popularity[12]. Although a greater focus on threatened species ex situ could result in greater in situ conservation, this may not influence visitor attendance, which is more important in determining overall in situ contributions.

The positive effects that total number of animals, mammal species richness and mean species body mass all have on attendance, together with the direct correlation between attendance and in situ project activity, supports the flagship approach of exhibiting large vertebrates. This indicates that institutions with numerous large-bodied species, and in particular mammals, are

more likely to achieve higher annual attendance and contribute to a greater number of in situ conservation projects. This provides the first indication, to our knowledge, that the flagship approach of using popular, large vertebrates in zoo collections to drive public education and in situ conservation fundraising is being utilised effectively to significantly increase the in situ conservation contributions of zoos globally. The flagship approach potentially results in increased global wildlife conservation, as greater financial in situ contributions in particular, have been shown to increase project success and viability[19].

Achieving a collection composed of numerous large-bodied species may encounter significant hurdles as demonstrated by the support for trade-offs revealed between increasing mean species body mass and both the total number of animals and collection dissimilarity. While the strong direct effect of mean species body mass on attendance provides support for the inclusion of large-bodied species, the trade-offs encountered with increasing mean species body mass present an alternative strategy—exhibiting numerous, unique, smaller-bodied species. These alternative correlative pathways influencing attendance and in situ project activity demonstrate that several alternative collection compositions can result in high attendance and in situ contributions, potentially resulting in the future diversification of collection planning strategies. Our results indicate the need to consider multiple direct and indirect drivers of attendance to enable the detection of trade-offs, and avoid collection planning and policy formation that do not take the full complexity of the system into account.

Increased concerns over the welfare of large vertebrates under human care can cause significant decreases in attendance[20], highlighting the importance of acquisition, welfare and management considerations. Our results indicate that ethical, management and welfare considerations may conflict with simplistic attendance maximisation strategies. For example, although collection dissimilarity is positively correlated with attendance; population management and conservation breeding recommendations encourage institutions to consolidate their collections to enhance management efficacy, resulting in higher uniformity of collections[1]. In addition, the recommendation to replace large vertebrates with numerous, unique, smaller-bodied species fails to address the serious challenges to the establishment of ex situ populations for species not presently maintained[13]. These issues are not easily resolved, and trade-offs will become more common as animal welfare standards and enclosure sizes increase[8].

The utilisation of animals under human care to educate the public and increase in situ conservation contributions is in line with The World Zoo and Aquarium Conservation Strategy, which states explicitly that animals held in zoos should "play a conservation role that benefits wild counterparts"[21]. This reflects the flagship and the One Plan conservation approaches, both of which ultimately contribute to Target 12 (conservation of species) of the United Nations Convention on Biological Diversity Aichi Biodiversity Targets[22]. Historically, personal preferences, availability and competition between institutions were the main determinants of collection composition[13]. Today collection composition decisions are largely shaped by individual institutions in consultation with both regional and international associations, for example the Taxon Advisory Groups of regional associations, such as the AZA[11,13]. The evidence presented here may be used to help guide policymakers and collection planners to promote not only direct conservation, but also visitor attendance and in situ contributions.

Our findings support the continued exhibition of popular, large-bodied species to drive attendance and in situ conservation activity, but not exclusively so, in agreement with previous recommendations[12,13]. The exhibition of large numbers of animals in collections that are dissimilar to other zoos is a viable alternative strategy. Each institution must make value-driven decisions regarding their collection composition in order to fulfil their institution-specific goals[9], and to ensure the genetic and demographic sustainability of the species within the global zoo network. However, consideration of public preferences and expectations of collection composition can result in greater attendance and increased in situ conservation contributions.

## Methods

**A priori meta-model**. SEM integrates multivariate relationships, testing both direct and indirect effects within a system[23]. SEM requires a strong theoretical and empirical knowledge of the study system to guide model specification and modification[23]. We conducted a literature review of the relationships between institutional attendance, zoo species composition and in situ contributions. Based on this prior theoretical knowledge and proposed causal relationships, we developed a hypothetical a priori meta-model[24,25] (Supplementary Fig. 1). This meta-model represents general relationships between multiple variables, while omitting statistical details[24]. A thorough description of both the prior theoretical knowledge and proposed causal relationships used to generate the a priori meta-model depicted in Supplementary Fig. 1 are explained in Supplementary Note 1. This hypothesised causal diagram was combined with available data to test the effects of species body mass on institutional attendance in the context of institutional compositional characteristics and socio-economic variables.

**Data**. All data generated or analysed during this study are included in Supplementary Data 1–3. Annual attendance figures and institutional area were obtained from the IZY[26]. In the absence of available revenue data, we use visitor attendance as a proxy of income to potentially fund in situ activities. Institutional vertebrate species holdings (mammalian, avian, reptilian and amphibian) were obtained from Species360[27]. Species360 is an international non-profit organisation that hosts and develops the Zoological Information Management System (ZIMS), the largest database of comprehensive and standardised information on >1100 zoo and aquarium collections globally. IZY and Species360 member institutions were cross-referenced and theme parks, aquariums and conservation/science centers removed to prevent potential biases, resulting in a sample size of 458 institutions in 58 countries (Supplementary Fig. 2). Safari parks and similar drive-through animal parks were treated the same as other institutions.

Both the IZY and ZIMS databases are based on submitted records from individual institutions. While these databases have not been subjected to editorial verification, potentially permitting differences in attendance calculations (e.g., exclusion of annual pass holders) or failure to update species holdings, they represent the only global databases of zoo attendance figures[26] and collection composition records (ZIMS). As a result, ZIMS is used by the IUCN, Convention on International Trade in Endangered Species (CITES), the Wildlife Trade Monitoring Network (TRAFFIC), United States Fish and Wildlife Service (USFWS) and Department for Environment, Food and Rural Affairs (DEFRA)[28].

Taxonomy and the status on the IUCN Red List of Threatened Species™ were standardised for the 4822 vertebrate species present using the 'taxize' package in the statistical program R[29,30]. Species richness, number of animals, taxonomic and IUCN Red List status representation, and both alpha and beta diversity indices were calculated using data from ZIMS species holdings (see Table 1 for variables list).

Species body mass was obtained from the Species Knowledge Index[31], which standardises data across 22 different global demographic databases. Species-level body mass information was available for 4214 species. Body mass for the remaining 608 species was inferred at the genus, family or order level using the same datasets. This allowed the mean species body mass of each institution to be calculated as shown in Eq. (1).

$$\overline{M} = \frac{\sum_{i=1}^{n} x_i m_i}{\sum_{i=1}^{n} x_i} \tag{1}$$

Where $\overline{M}$ is the mean abundance weighted species body mass per institution, $x_i$ is the number of individuals of species $i$, $m_i$ is the body mass of species $i$, where $i$ goes from 1 to $n$ species per institution.

To assess socio-economic factors, we used GDP and national population size for each country[32]. Institutional GPS co-ordinates were used to calculate total population sizes within 10 km radii in ArcGIS using estimated global population counts[33].

In order to assess the in situ contributions of individual institutions the AZA Annual Report on Conservation and Science was consulted[34]. This provided the number of field conservation programmes, in which AZA member institutions were involved in 2015. When cross-referenced with IZY and Species360 members, this provided a sample size of 119 institutions across four countries for which we could analyse in situ contributions. The number of projects, as a measure of in situ conservation contributions, does not provide further resolution on the form the contribution takes (e.g., financial, expertise, resources, animals, training, etc.). However, a separate analysis of the relationship between the number of in situ

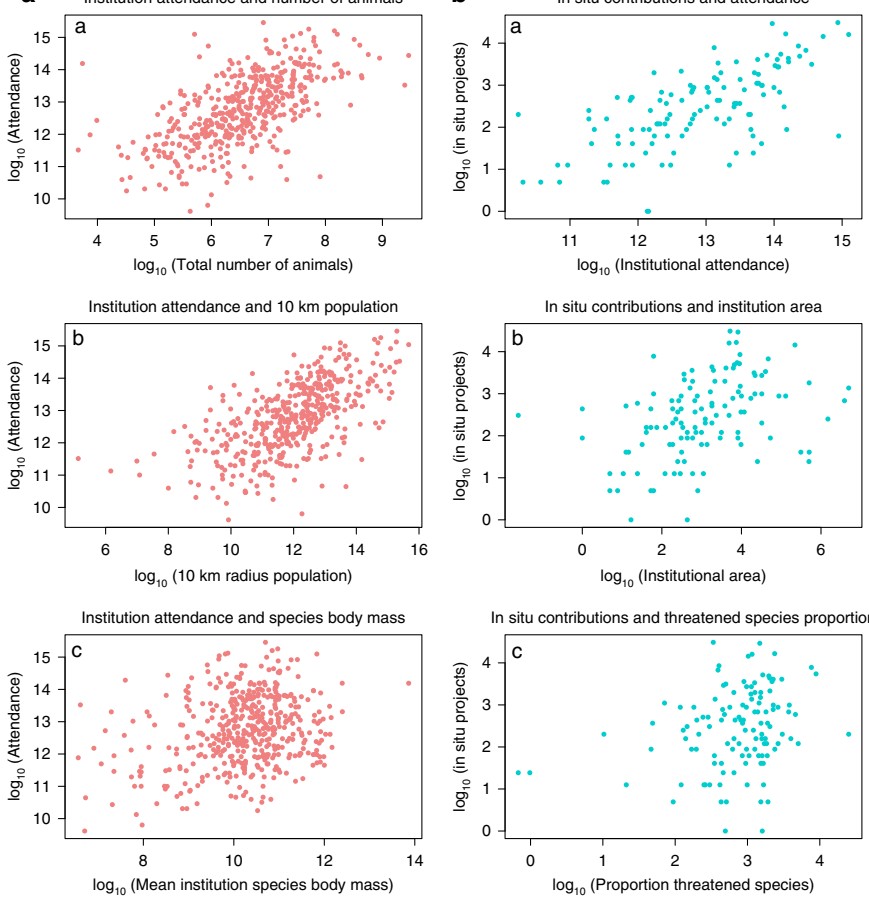

**Fig. 3 Bivariate relationships between institutional attendance, in situ contributions and their strongest predictors. a** (left panel, $n = 458$), $log_{10}$ transformed bivariate plots of institutional attendance and total number of animals, 10 km radius population and mean species body mass a–c. **b** (right panel, $n = 119$), $log_{10}$ transformed bivariate plots of institutional in situ contributions and attendance, institutional area and the proportion of threatened species present per institution a–c. All variables are adjusted for species abundance per institution. Source Data: Supplementary Data 1 and 2 provided.

projects supported and the total in situ financial investment per institution was conducted on anonymised data from 83 individual British and Irish Association of Zoos and Aquariums (BIAZA) institutions. These data show a clear positive relationship between the number of in situ projects supported and total in situ financial expenditure. As this data set was anonymised, we were unable to include it in our integrated model; however, these data are shown in Supplementary Fig. 5 and support our assumption that the number of in situ projects is a meaningful proxy for the total in situ financial investment per institution.

**Analyses**. Two distinct SEM frameworks were tested, the attendance model and the in situ model. The attendance model tested the relationship between visitor attendance and all the various specified variables for 458 institutions globally. This model did not include any in situ contribution data. The in situ model tested the relationship between visitor attendance, in situ contributions and all the various specified variables for a subset of 119 institutions in North America for which in situ contribution data were available. The results of the attendance model were used to guide the development of the attendance linked pathways in the in situ model as the larger sample size of the attendance model had higher power. The results of the attendance model are combined with the results of the in situ model in Fig. 2, with a yellow box delineating the boundary of the two models. Only the additional in situ pathways of the in situ model are reported, as all other relationships were derived from the attendance model due to its higher statistical power.

All analyses were carried out using the R program (version 3.4.3) and the packages 'lavaan'[35] and 'lavaan.survey'[36] for SEM. All variables were both mean centered and expressed in units of standard deviation to allow direct comparisons of effect sizes between variables.

**Attendance model**. We combined prior theoretical knowledge and proposed causal relationships to create the a priori SEM meta-model (Supplementary Fig. 1 and Supplementary Note. 1). The meta-model captured all evidence-based relationships that we found in our literature review and all plausible and suspected predictors of attendance that we hypothesised. This model was then

refined to create the final model depicted in Fig. 2 using the approach described in Grace et al.[37] and similar to that implemented in Grace et al.[25]. In summary, the a priori meta-model was modified through addition and deletion of pathways using model-data fit procedures to produce a range of plausible alternative models which were compared using corrected Akaike's Information Criterion (AICc) values. All modifications to the model, with pathways removed or inserted, were based on quantitative recommendations, theoretical intuition and model-data fit. Model-data fit was assessed using a combination of absolute fit indices (e.g., Standardised Root Mean Square Residual) and incremental fit indices (e.g., Comparative Fit Index), to account for the differential sensitivity of fit indices to data distribution, model size and sample size[38]. Modification indices were used to guide the addition of suspected pathways, with a standard cut-off level for the chi-square test criterion of 3.84[39]. Highest value modification indices were considered first, however, as modification indices do not take into account whether or not relationships make theoretical sense, intuitive theoretical relationships were also considered, as outlined in the Supplementary Code provided. Following the addition of these pathways, p values were then used to identify potentially unsupported pathways, with a threshold of 0.05. Highest p values were considered first for removal. Overall model selection from the pool of competing models was achieved using AICc values[40], with a threshold of more than two AICc units lower than the nearest competing model being considered sufficient for model selection. The AICc values of competing models are shown in Supplementary Table. 1. The final selected attendance model was validated, using four random subsets of the existing data ($n = 200$ each time), to ensure parameter estimates were similar when using different datasets from the same sample[23]. Institutions were included within countries in the model.

**In situ model**. Due to the lower sample size in the in situ model, which only covered four countries, we did not include GDP and country as variables. We started with the most complete model to predict both in situ contributions and attendance. Initial attendance links were based on the results of the best attendance

model. Model-data fit and model selection were assessed in the same manner as for the attendance model.

Tests of mediation were performed on mediated pathways to ensure both direct and indirect effects of variables were justified in both models. Values for both Absolute Fit Indices and Incremental Fit indices were supportive of good model fit (Supplementary Table. 2). All standardised path coefficients, total effect sizes, significance values and proposed interpretations of causal pathways for both models are shown in Table 2. See Fig. 3 for bivariate relationships between attendance, in situ contributions and their strongest predictors. These models incorporate species abundance per institution, however, models using species presence–absence only were also assessed and provided overall similar results and conclusions, with qualitative differences found in only four links per model (see below). An updated meta-model reinforces many previously supported relationships, such as those between species body mass, species richness and the number of animals present (Supplementary Fig. 3).

**Species presence–absence SEM frameworks**. The attendance and in situ model results reflecting species presence–absence only are shown in Supplementary Fig. 4. Chi-squared statistics, fit indices, standardised path coefficients and proposed interpretations for both the attendance and in situ models reflecting species presence–absence are also presented (Supplementary Tables 3 and 4). Residual covariances for both the attendance and in situ models are shown in Supplementary Table. 5 (species abundance models) and Supplementary Table. 6 (species presence–absence models).

**Reporting summary**. Further information on research design is available in the Nature Research Reporting Summary linked to this article.

## Data availability

The anonymised data required for both the attendance and in situ SEM and the source data for Supplementary Fig. 5 are provided as Supplementary Data files with associated metadata in the description of additional Supplementary Files. We provide the four subsets of the Attendance SEM data used to validate the models depicted in Fig. 2 and Supplementary Fig. 4 as Supplementary Data 4–7.

## Code availability

A compiled R-markdown pdf including code and commentary is provided as Supplementary Code. The uncompiled R-markdown code file to reproduce the SEM analyses is available from https://github.com/yvonnebuckley/Zoo-attendance.

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

## Acknowledgements

We acknowledge and thank the >1100 Species360 member institutions for their continued support and data input. Additionally, we would like to thank AZA and BIAZA for their contribution of in situ contribution data to this paper. We thank BIAZA for providing the data for Supplementary Fig. 5. This research was funded by the Irish Research Council Laureate Awards 2017/2018 IRCLA/2017/60 to Y.M.B. In addition, K.H. was funded by the Marie Sklodowska Curie Research Grants Scheme, grant 749594 and D.A.C. was funded by Species360 and the University of Southern Denmark.

## Author contributions

All authors developed the concept of the manuscript. A.M., D.C. and K.H. collected the data. A.M. undertook the analysis and drafted the text in consultation with Y.M.B. A.M. produced the tables and figures. All authors contributed to the writing of the MS.

## Competing interests

The authors declare no competing interests.
