## [Peer Review File · Nature Communications]

Reviewers' comments:

Reviewer #1 (Remarks to the Author):

General points:

This study provides invaluable data and important results with implications for zoo collection planning. The relationship between animal collection composition, attendance and contribution to conservation represents a major gap in examining the role of zoos, and this study represents an important first step. I would like to see additional / alternative data analysed in the in situ component of the model but I would still recommend publication if the authors provide justification as to why this would not be feasible.

In situ contributions: a major consideration for revising this study would be the nature and extent of the in situ contribution data. As I'm sure the authors are aware, the number of projects supported is not the best metric for in situ conservation support, primarily because zoos may choose to allocate all their efforts in one or a very small number of projects rather than many. Given the difficulty of measuring actual impacts on wild populations and ecosystems, financial investment may be a more direct measure of zoo in situ contributions. Funding contributions are reported annually as a condition of membership to the British and Irish (BIAZA), European (EAZA) and North American (AZA) zoo associations – would it be possible to access this data? The number of projects would also be available from these sources to cross-reference with the attendance dataset.

At the very least there should be a sentence in the text acknowledging that number of projects has limitations as a measure of conservation investment, in addition to the geographic and sample size limitations already mentioned. As a minor point, the wording "in situ contributions" suggests financial contributions and could potentially mislead the casual reader. More direct use of "number of in situ conservation projects" or potentially "in situ conservation activity" for shorthand (as in the title) would help avoid confusion.

Model selection: selection of the final models using AICc should be specified more explicitly. We do not know as the reader whether the final models were very clearly the best fit to the data or only narrowly so (and therefore potentially equivalent to other combinations of terms or pathways). I would like to see a statement of how this decision was made (e.g. more than 2 AIC units lower than nearest competing model) and a table of potential best model structures in the extended data if relevant.

Additional diagram suggestion: would it be possible to illustrate hypothetical collection compositions which the model indicates would maximize attendance levels, e.g. certain proportions of small or large-bodied species if other variables are held constant? As this is a main conclusion of the study it would be nice to illustrate it directly in addition to the current figures.

Reference numbering: there appears to be a few references which do not match the citation numbers? I suspect that reference to 14 (Bowkett) should in some cases be 13 (Hutchins et al.)? This may be the case for the main text but seems to be affecting the meta-model diagrams (Extended Data Figures 1 and 4).

Specific comments:

Title: should the title refer to attendance? The more robust findings pertain to visitor numbers rather than conservation support, maybe both should be mentioned?

L27-28: The trends documented by Balmford et al. relating to cost, breeding and enclosure size are valid but there is noise in the data, i.e. some large-bodied mammals breed consistently in captivity.

This sentence should be amended to be less absolute and refer to a comparator e.g. "Compared to smaller species, large animals are often costlier to maintain, more difficult to breed in captivity..."

L52: what does "popularity of institutional collections" refer to exactly?

L51-54: Is there evidence for this flagship effect? I'm not sure there is anything more than a few examples in the articles cited (13 and 14). Rephrase to explain the concept without reference to evidence.

L65: As explained above, the phrase in situ contributions is potentially misleading.

L73: "to more" rather than "more to"

L86-87: Why was an effect of threatened species proportion on attendance expected? The link in Extended Data Figure 1 is not labelled with a citation.

L108-109: Reference 14 doesn't provide evidence of perceived species popularity. Perhaps a mis-reference?

L143-144: Suggest “This is because” rather than “This indicates that”.

L146-149: This conclusion should be tempered slightly. We do not know that greater numbers of in situ projects leads to greater conservation impact (projects may fail or even have negative impacts), reword to indicate an impact on in situ investment or similar. The last part of the sentence: “as greater in situ contributions have been shown to increase project success and viability” also needs clarifying. Presumably project success is increased through greater financial contributions but that is not what has been measured in this study.

L180-181: Correct reference? Possibly refers to Hutchins et al. 1995. Suggest change to “Historically, personal preferences, availability, and competition between institutions were the main determinants of collection composition”.

L182: Suggest “decisions are also shaped by...” or “largely shaped”.

L190: Reference 13?

L218: Would be interesting to know how safari parks and similar drive-through animal parks were classified.

Andrew E Bowkett

28th May 2019.

Reviewer #2 (Remarks to the Author):

This is a well written paper that describes a significant contribution to an issue that really demands a critical analysis. Such a critical analysis is challenging because the issue of maintaining animals in zoos and the potential of zoos to contribute to conservation, whether in situ or ex situ, or surrounded by very strong views, both for and against. At the same time, however, there is a real need for marshalling resources for in situ conservation, as is clear from both the recent IPBES Global Assessment and will be clear for species particularly in the reporting for the present Convention on Biological Diversity Strategic Plan, which concludes in 2020.

My comments are rather minor, but would, I think help with increasing confidence in the framing and design on the study. First, although there is much written about various aspects of zoos and captive management, little is in what might be considered front line journals. There is therefore quite a diversity of sources cited in the article and they are presented as sources of equivalent quality and merit. For example, an approach that proposed in the WAZA Magazine is cited, whereas the IUCN ex situ management guidelines (with a supporting publication in Conservation Letters) is not. I am not entirely sure of the veracity of some of the references to zoo species journals and would suggest that the authors consider the extent to which they really do have confidence in them.

Second, there is the option, although I accept word limits, to briefly emphasise the policy context. The EU Zoos Directive has been through a REFIT evaluation, IUCN has relatively new guidelines, and CBD has a clear focus on species conservation (Target 12) that was deemed to be 'moving away from the target' at the mid-term review of the present Strategic Plan in 2014. All of which pleads for the analysis presented here.

Finally, it would have been helpful to see a more discursive approach to defining the model. This is too easily dispatched in lines 204-205 and so does not inspire confidence that the literature review has led to a defensible model. The pathways are stated but not justified and so there is a need to have a process posited for each pathway. Without this it could be argued that it just becomes an exercise in finding the best SEM across a big square correlation matrix. For example:

we assumed that visits to zoos would be positively associated with numbers of visits to zoos because big animals are more charismatic and are likely to attract more visitors (as an example). Then, for example, Smith (1990) demonstrated that size matters.

This would help show that the model is not based on a series of one off relationships but the study of the whole system. Explicitly justifying each of the single variable relationships helps make clear the justification for the creation of the model. The effect on the non-SEM specialist reader, like me, would be to show that the model pathway is based on an explicitly defined underlying process.

Reviewer #3 (Remarks to the Author):

This is an interesting and generally well-written paper that utilizes a large data set to assess the relationships between zoo 'size' (= area and number of visitors) and various variables linked to the role of zoos in conservation. The relationships are explored using structured equation modelling, which I am not familiar with, but the links between the variables are nicely displayed. I enjoyed reading the paper and believe that there is information presented here that will be of some interest to the zoo community and those involved with ex-situ conservation. However, despite the sophisticated analysis I wonder if there is anything that is really new here? My guess is that the main finding that bigger zoos are more involved with in-situ conservation will come as no surprise to the zoo community. Indeed, I suspect that similar relationships would emerge with almost any business, i.e. bigger businesses are likely to invest more in charitable and corporate social responsibility activities. A secondary result that apparently emerges is that an alternative strategy would be for zoos to focus on more, unique smaller species, but I can't see where this result emerges from the results that are presented. These points are elaborated on in some of the specific comments below.

Specific queries:

Lines 34-36: This is the key result. But is it really surprising to find that zoos with many large animals get more visitors and contribute to more in-situ conservation projects?

Line 50: It is rather over-simplistic to suggest that in-situ conservation activities are funded through paying visitors. There are various models by which such activities are funded, e.g. external grants, conservation campaigns, special events, zoo memberships, entrepreneurial activity etc.

Lines 52-54: Needs slight rewording, as it suggests that 'Strategic Collection Planning' is about large vertebrates.

Lines 58-59: I'm not entirely sure what the comment about 'bivariate relationships' is referring to. Presumably, it is trying to suggest that multiple variables need to be considered together when exploring these relationships? Perhaps tighten up.

Line 73: A slight rewording could tighten this up, i.e. 'We found that zoos with high attendance contribute to more in situ conservation projects'.

Lines 76-78: Although tangential to the main point of the paper, a qualifier here is that socio-economic variables may play a lesser role in influencing attendance if admission to the zoo is free or highly subsidised. This may be the case in countries where zoos are run by national or local government.

Lines 88-89: I can't follow this sentence as the first part appears to contradict the second part.

Lines 97-100: I think a little more caution is needed here about cause and effect. The analyses show relationships, but the old adage that correlation does not equal causation needs to be considered. There may be some subtle nuances within the data that are playing roles here.

Lines 104-105: Again, I find the result that greater investment in ex situ conservation is positively related to in situ conservation not very surprising. Indeed, this is the direction that zoos have been heading in for some years, and is enshrined within the World Zoo Conservation Strategy. There are many good existing case studies of this within the literature.

Figure 1 (and generally for the methods): I am not familiar with the SEM framework for analysing such relationships, but I did wonder what the advantages might be over a traditional GLMM?

Table 1: 'Number of in situ projects' is used as the variable to measure zoo involvement in field conservation. Although some relationships with this variable apparently emerge, this is a fairly coarse measure of in situ activity. Zoos have different models for investing in in situ conservation: some support a wide range of projects with modest funds, while others invest more substantially in a smaller number of projects. So my points are that (1) some relationships may be masked by using 'number of projects'; and (2) conservation spend on in situ projects may actually reveal other relationships.

Lines 146-147: Implies that Strategic Collection Planning is all about flagship species – needs rewording to take account of the fact that it is just one element.

Lines 156-157: The alternative strategy of exhibiting numerous, unique, smaller-bodied species emerges here as another key finding, but I cannot see where this is explicitly produced from the results that are presented earlier.

Lines 165-166: Again, I cannot see where the results show this conflict. This seems to be a speculative inference of the results perhaps, rather than a result in itself.

Lines 166-170: This implies that there is a trade-off between collections avoiding similarity to avoid competition for paying visitors, while breeding programmes encourage collections to consolidate collections on similar species as this makes species management more efficient. This is probably true to some extent, but again there are subtle nuances at play here. Many breeding programmes are managed on a European-wide scale (e.g. via EAZA) and it is unlikely that, for example, zoos in the UK and Germany would compete with each other because they hold the same species. Likewise, on a regional scale, there is already clear dissimilarity on regional scales. For example, East Anglia is an area with a historically high density of zoos, and back in the 1980s there were major collections within about 50 miles of each other focusing on Europe, Latin America and Asia. Although some of these collections no longer exist the specialisms within the regions persist. So in sum, I think this trade-off is not really a big issue when it comes to collection planning because there are more important drivers at play here.

Lines 184-186: Overall, I think this is rather an over-statement of the relevance of the results to policy-makers, collection planners and conservation. The results will certainly be of interest, but the bigger zoos have marketing teams that have a strong handle on what drives visitor attendance and conservation teams that appraise what are the most cost-effective in-situ projects to get involved with given zoo income streams and strategic objectives.

Lines 190-191: As mentioned above, I think this is already being done on a regional scale. Witness the differences between collections in areas such as East Anglia and Kent.

Lines 218-219: Removal of aquariums from the data set is perfectly justified, but what about other specialist zoos, such as bird parks and those with a geographical specialism?

Reviewer #4 (Remarks to the Author):

I was asked to evaluate the methodology of this paper. As I am by no means an expert on zoos or conservation planning, I will not comment on content unrelated to the methodology. That being said, I have several concerns regarding the way the Structural Equation Models were performed. See my comments below.

- Lines 76-82: "Collection composition variables (no. of animals, species body mass, compositional dissimilarity and mammal species richness) are more important in determining attendance than socio-economic variables."

I am not so sure about this. Given the many indirect pathways linking attendance to collection composition and socio-economic variables, it would be quite helpful to use the standardized path coefficients to calculate total effects. In this way, you could for example calculate the total effect of mean species body mass on attendance, which is here the sum of:

- the direct effect on attendance
- the indirect effect mediated by dissimilarity
- the indirect effect via species richness and dissimilarity
- the indirect effect via species richness and total animals
- the indirect effect via species richness, total animals and dissimilarity

Of course, this ignores pathways via correlated exogenous variables, so I would also include covariances among exogenous variables, unless these were fixed, which is not clear to me (see also next comment), and which I would not recommend unless there are strong theoretical reasons to assume these are truly uncorrelated.

The point is: it may well be that the total effect of collection composition variables on attendance is weaker than that of socio-economic variables.

"The total number of animals had the largest direct positive effect on attendance, followed by abundance weighted mean species body mass, with compositional dissimilarity and mammal species richness having smaller direct positive effects (Fig. 1). Ultimately, there is strong evidence that the presence of larger-bodied species is strongly related to increased institutional attendance and in turn, in situ contributions."

I have two issues with this. First, the second-largest direct positive effect on attendance is that of "10 km population", not that of "mean species body mass". Second, mean species body mass has strong negative effects on dissimilarity and species richness, both of which increase attendance. So again, without quantifying these indirect effects, I am not sure whether mean body mass is related to increased institutional attendance.

- Line 89-93: "Mammal species richness had a direct positive effect on attendance as well as an indirect positive effect through the total number of animals. Mammal species richness could have a small negative effect on attendance if the increase in total species richness does not lead to increased total number of animals."

1) The indirect positive effect is not only through the total number of animals, but also through the total species richness, which positively influences total number of animals. Moreover, there could be an indirect negative effect of mammal species richness on attendance by increasing total species richness, which in turn has a direct negative relationship with attendance.

2) The increase in total species richness does lead to increased total numbers of animals, based on your model, so I do not understand the meaning of the second sentence here.

- Line 95: "No support was found for linking species body mass directly with in situ project activity". But this pathway is not present in the model? Or did you only depict significant pathways? I would suggest to display non-significant paths, for example with dashed arrows.

- Line 100. Or by having a higher mammal or total species richness, presuming that their total effect on attendance is positive (which you could calculate).

- Line 138: "See Extended Data Tables 1 and 2 for test statistics and fit indices, standardized path coefficients, significance values and proposed interpretations of causal pathways." This is a bit misleading, as it suggested to me that the path coefficients shown are unstandardized, and the standardized ones can be found in supplementary data. However, they match, so all values are standardized. I think it would be nice to provide unstandardized coefficients as well, in case people want to use your SEM for things like predictions. It would also be nice to see the actual P-values rather than the message that they all were below 0.05. There is a big difference between a 0.049 and a 0.001. I would also like to see R² values for all endogenous variables, and residual variances. Were residual covariances allowed in the model, and if so, what were their values? Were means, variances and covariances of exogenous variables free or fixed?

- Line 157. "no single optimal collection composition exists". Based on your model, you could calculate which of the two proposed strategies increases in situ conservation investment most, instead of speculating about it.

- Line 201. What makes you think that the multivariate relationships in SEM are causal? SEM is not different from any other form of data analysis, in that it cannot replace comparison of experimental manipulation with controls, which is the only way to establish causality. SEM based purely on observed, unmanipulated variables therefore can never prove causality.

- Line 204. The explanation about the meta-model is a bit vague.

- Line 258. Analyses. Several aspects of the used analyses are unclear to me. You build an a priori meta-model (whatever that means), represented in Extended Data Figure 1. I guess this is the theoretical model based on prior knowledge. However, it is very unclear how you went from this model to the final model represented in Figure 1. The caption of Extended Data Figure 4 states: "grey dotted lines indicate relationships that were not directly assessed, but incorporated into analyses" What does this mean? How could they be incorporated into analyses without being assessed? There are variables in this model that do not occur in the final model and vice versa, or is "collection taxonomy" the same as "dissimilarity"? Or are these latent variables with multiple indicators? Supplementary Tables 1 and 2 lists the variables for Attendance and In Situ models, but variables such as "50km_pop" and "Mean.Jaccard.Dissim.P.A." do not occur in any of the models you present in this manuscript? The only clue the reader gets is that "Modifications to the model with pathways removed or inserted were based on model-data fit. All modifications were made based on logical interpretations of the data resulting in the elimination of certain variables." What was the exact basis for doing so? Modification indices? What were the cutoff levels? In principle, each modification you make to a hypothetical model of which the overall fit fails upon testing, requires new, independent data. Results from a modified model must be considered provisional only. Adding and removing paths until the overall model fit is adequate, or all paths become significant, are clear examples of overfitting. I have the impression that all "non-supported" links were just the non-significant ones, and subsequently removed from the a priori model, while this is not warranted. Line 293 states that significance values can be found in Extended Data Table 2, but I do not see them there. The explanation of how the Attendance model was linked to the In Situ model (lines 285 to 289) is also not at all clear to me. In general, I would like to see these things spelled out in more detail. Especially useful would be to have the R code of all the different steps and analyses available in a supplementary file. Then at least the reader could get a better grasp of the whole procedure.

- Line 268. By centering and standardizing, the path coefficients become useless for prediction. Why not just report both unstandardized and standardized coefficients?

- Line 297. "however models using species presence-absence only were not qualitatively different". I see some clear differences between Fig. 1 and Extended Data Fig. 5. For example, in the latter there is a strong positive link between total animals and dissimilarity and there is a negative link between species richness and dissimilarity, both of which are absent in Fig. 1, and there are other differences. These are clearly qualitative differences.

Mooney et al. Response to Review

We thank the editor and the reviewers for their input which has greatly improved our manuscript.

We have updated our manuscript based on the editor's and reviewers' suggestions and respond to each of the reviewers' comments below in *italics*. We have highlighted major changes to the MS file in yellow and have added line numbers to aid comparison between this response document and the MS.

We have numbered the reviewers' comments throughout in order to cross-reference them when necessary. We have added a 150 word abstract to the manuscript in order to comply with Nature Communications format.

Reviewer #1 (Remarks to the Author):

General points:

1.01 Comment: This study provides invaluable data and important results with implications for zoo collection planning. The relationship between animal collection composition, attendance and contribution to conservation represents a major gap in examining the role of zoos, and this study represents an important first step.

1.01 Response: *We thank reviewer #1 (Dr. Andrew Bowkett) for his encouraging assessment.*

1.02 Comment: I would like to see additional / alternative data analysed in the in situ component of the model but I would still recommend publication if the authors provide justification as to why this would not be feasible. In situ contributions: a major consideration for revising this study would be the nature and extent of the in situ contribution data. As I'm sure the authors are aware, the number of projects supported is not the best metric for in situ conservation support, primarily because zoos may choose to allocate all their efforts in one or a very small number of projects rather than many. Given the difficulty of measuring actual impacts on wild populations and ecosystems, financial investment may be a more direct measure of zoo in situ contributions. Funding contributions are reported annually as a condition of membership to the British and Irish (BIAZA), European (EAZA) and North American (AZA) zoo associations – would it be possible to access this data? The number of projects would also be available from these sources to cross-reference with the attendance dataset.

1.02 Response: *We agree that the in situ data (number of projects) presented here is a proxy for conservation effort and that a total financial contribution per institution would also be useful. As a result, we have contacted and sent data/collaboration requests to AZA, EAZA, BIAZA, ALPZA (South America), ZAA (Australia) and PAZA (Africa) and invited them to contribute data and expertise to this paper. Specifically, we requested the number of in situ projects and if possible, the total annual in situ financial contribution per institution. Unfortunately, given the sensitive nature of the data, our requests for collaboration have yet not been granted, with no indication of a timeline within which we would be able to compile an adequate data-set from these*

multiple sources. However, we were able to obtain additional anonymised data from BIAZA. These data show the relationship between the number of in situ conservation projects supported and the total financial in situ conservation expenditure for 83 individual BIAZA institutions for the year 2018. These data show a clear positive relationship between financial expenditure and the number of in situ projects supported. As these data were anonymised, we were unable to include them in our modelling procedure, however these data and their relationship are now shown in the new Supplementary Figure 6. These additional data justify our assumption that the number of in situ projects is a meaningful proxy for the total in situ financial investment per institution. The AZA data already presented in the main manuscript is the only known standardised, publicly available in situ contribution data. Therefore, we thank the reviewer for still recommending publication in the absence of further data availability.

1.03 Comment: At the very least there should be a sentence in the text acknowledging that number of projects has limitations as a measure of conservation investment, in addition to the geographic and sample size limitations already mentioned.

1.03 Response: *We agree that this needs to be noted, and we have added the following paragraph to make this point clear and to include the additional BIAZA data obtained:*

“The number of projects, as a measure of in situ conservation contributions, does not provide further resolution on the form the contribution takes (e.g. financial, expertise, resources, animals, training etc.). However, a separate analysis of the relationship between the number of in situ projects supported and the total in situ financial investment per institution was conducted on anonymised data from 83 individual British and Irish Association of Zoos and Aquariums (BIAZA) institutions. These data show a clear positive relationship between the number of in situ projects supported and total in situ financial expenditure. As this data-set was anonymised, we were unable to include it in our integrated model; however, these data are shown in Figure 3 and support our assumption that the number of in situ projects is a meaningful proxy for the total in situ financial investment per institution” (Lines 299 – 309).

1.04 Comment: As a minor point, the wording “in situ contributions” suggests financial contributions and could potentially mislead the casual reader. More direct use of “number of in situ conservation projects” or potentially “in situ conservation activity” for shorthand (as in the title) would help avoid confusion.

1.04 Response: *Given the diversity of possible in situ contributions (financial, expertise, resources, animals, training etc.) we do not believe that our wording suggests financial contributions. Although we assessed the number of in situ conservation projects per institution, the actual contribution made by each institution to each project could take several forms, however, we lack the data to delineate the form the contribution takes, so we chose to keep the term as broad as possible. We believe that “in situ conservation activity” suggests direct ‘hands-on’ involvement, whereas the contribution could simply be financial. Therefore, we believe that our original wording should be retained. We clarify this point in Table 1, where In Situ Contributions is defined as “The annual number of field conservation programmes in which individual AZA member institutions contribute to in some capacity (2015)”.*

1.05 Comment: Model selection: selection of the final models using AICc should be specified more explicitly. We do not know as the reader whether the final models were very clearly the best fit to the data or only narrowly so (and therefore potentially equivalent to other combinations of terms or pathways). I would like to see a statement of how this decision was made (e.g. more than 2 AIC units lower than nearest competing model) and a table of potential best model structures in the extended data if relevant.

1.05 Response: *We have updated this paragraph to better describe how we created our original model and chose our final model:*

“We combined prior theoretical knowledge and proposed causal relationships to create the a priori SEM meta-model (Supplementary Figure 1 and Supplementary Methods). The meta-model captured all evidence-based relationships that we found in our literature review and all plausible and suspected predictors of attendance that we hypothesised. This model was then refined to create the final model depicted in Figure 2 using the approach described in ³⁷ and similar to that implemented in ²⁵. In summary, the a priori meta-model was modified through addition and deletion of pathways using model-data fit procedures to produce a range of plausible alternative models which were compared using AICc values. All modifications to the model, with pathways removed or inserted, were based on quantitative recommendations, theoretical intuition and model-data fit. Model-data fit was assessed using a combination of absolute fit indices (e.g. Standardised Root Mean Square Residual) and incremental fit indices (e.g. Comparative Fit Index), to account for the differential sensitivity of fit indices to data distribution, model size and sample size³⁸. Modification indices were used to guide the addition of suspected pathways, with a standard cut-off level for the chi-square test criterion of 3.84³⁹. Highest value modification indices were considered first, however as modification indices do not take into account whether or not relationships make theoretical sense, intuitive theoretical relationships were also considered, as outlined in the Supplementary R Code provided. Following the addition of these pathways, p-values were then used to identify potentially unsupported pathways, with a threshold of 0.05. Highest p-values were considered first for removal. Overall model selection from the pool of competing models was achieved using AICc values⁴⁰, with a threshold of more than 2 AICc units lower than nearest competing model being considered sufficient for model selection. The AICc values of competing models are shown in Supplementary Table 1. The final selected Attendance model was validated, using four random subsets of the existing data (n = 200 each time), to ensure parameter estimates were similar when using different datasets from the same sample²³. Institutions were included as a nested variable within countries in the model” (Lines 331-356).

We hope this clarifies the model selection process.

1.06 Comment: Additional diagram suggestion: would it be possible to illustrate hypothetical collection compositions which the model indicates would maximize attendance levels, e.g. certain proportions of small or large-bodied species if other variables are held constant? As this is a main conclusion of the study it would be nice to illustrate it directly in addition to the current figures.

1.06 Response: *Our SEM results do not provide a “prescription” for maximising attendance and in situ conservation project contributions as this can be achieved through several potential collection compositions. Indeed, the multiplicity of compositions that can be used to drive attendance is one of the main conclusions of the manuscript. We feel it would be somewhat misleading to present a very limited number of compositions in a figure. Furthermore, without a thorough comprehension of the economic, logistical and regulatory constraints involved in changing composition it is not possible to suggest ‘optimal’ compositions. In addition, diminishing returns are not explicitly modelled in our SEM procedures, therefore such an analysis would not be statistically defensible. We hope that the inclusion of total effect sizes in response to Reviewer comment 4.01 helps to clarify the strongest compositional drivers of attendance and in situ contributions. We display this information graphically in Figure 1, with a simplified version of our SEM, depicting only the total effect sizes. Please see Response 4.13 for further discussion of this topic.*

1.07 Comment: Reference numbering: there appears to be a few references which do not match the citation numbers? I suspect that reference to 14 (Bowkett) should in some cases be 13 (Hutchins et al.)? This may be the case for the main text but seems to be affecting the meta-model diagrams (Extended Data Figures 1 and 4).

1.07 Response: *We thank the reviewer for pointing this out and apologise for the error, the mismatch was between reference 14 (Bowkett, 2014) and reference 12 (Whitworth, 2012). These have been verified and amended as necessary throughout the text. The meta-model diagrams (Supplementary Figures 1 and 4) have also been changed accordingly.*

Specific comments:

1.08 Comment: Title: should the title refer to attendance? The more robust findings pertain to visitor numbers rather than conservation support, maybe both should be mentioned?

1.08 Response: *We have now changed the title to “Few large animals or many small? Managing zoo collections for visitor attendance and in situ conservation” as suggested.*

1.09 Comment: L27-28: The trends documented by Balmford et al. relating to cost, breeding and enclosure size are valid but there is noise in the data, i.e. some large-bodied mammals breed consistently in captivity. This sentence should be amended to be less absolute and refer to a comparator e.g. "Compared to smaller species, large animals are often costlier to maintain, more difficult to breed in captivity..."

1.09 Response: *Changed to “However, compared to smaller species, large animals are often costlier to maintain, prove more difficult to breed in captivity, require larger enclosure sizes⁵ and raise ethical and welfare issues⁶” (Lines 37-39) as suggested.*

1.10 Comment: L52: what does “popularity of institutional collections” refer to exactly?

1.10 Response: *In the study cited (Whitworth, 2012), the popularity of institutional collections refers to the types and numbers of species kept, as a result we have changed the text to “the popularity of institutional collections (in terms of the species within the collection)...” (Lines 50 – 51).*

1.11 Comment: L51-54: Is there evidence for this flagship effect? I'm not sure there is anything more than a few examples in the articles cited (13 and 14). Rephrase to *explain the concept without reference to evidence.*

1.11 Response: *We believe that the articles cited not only provide a very good overview of what the flagship effect is, but that the examples cited also provide a unique insight into how the flagship effect can be implemented effectively. We believe there is evidence of the flagship effect, for example upon opening its "Congo Gorilla Forest" in 1999, the Bronx Zoo not only imposed a special admission fee to support wildlife conservation in tropical African forests, but also allowed visitors to choose how their fees could be spent in situ. By 2009, \$10.6 million had been raised and expended on African forest wildlife conservation from this source alone (Conway, 2011). We have amended the references to include this specific example of how the flagship effect can be implemented effectively.*

1.12 Comment: L65: As explained above, the phrase in situ contributions is potentially misleading.

1.12 Response: *see Response 1.04 above.*

1.13 Comment: L73: "to more" rather than "more to"

1.13 Response: *Changed to "to more" as suggested.*

1.14 Comment: L86-87: Why was an effect of threatened species proportion on attendance expected? The link in Extended Data Figure 1 is not labelled with a citation.

1.14 Response: *An effect of threatened species proportion on attendance was expected as Whitworth (2012) has previously shown that "rare" species are more popular than "common" species. This same study also found that the popularity of institutional collections (in terms of the species within the collection) is positively correlated with attendance. Therefore, we expect that the proportion of threatened species in a collection will be correlated with institutional attendance. This reference has also been added to Supplementary Figures 1 and 4 as suggested.*

1.15 Comment: L108-109: Reference 14 doesn't provide evidence of perceived species popularity. Perhaps a mis-reference?

1.15 Response: *This was mislabelled and has now been changed to Reference 12 (Whitworth, 2012).*

1.16 Comment: L143-144: Suggest "This is because" rather than "This indicates that".

1.16 Response: *As this was not an experimental study we cannot say with certainty that "This is because...", although this is indicated by the results. Therefore, in a bid to avoid statements of causality, we think it is best to retain the "This indicates that".*

1.17 Comment: **L146-149:** This conclusion should be tempered slightly. We do not know that greater numbers of in situ projects leads to greater conservation impact (projects may fail or even have negative impacts), reword to indicate an impact on in situ investment or similar.

1.17 Response: *This has been changed to temper the causality as follows: “This provides the first indication that the ‘flagship’ approach of using popular, large vertebrates in zoo collections to drive public education and in situ conservation fundraising is being utilised effectively to significantly increase the in situ conservation contributions of zoos globally. Potentially resulting in increased global wildlife conservation, as greater financial in situ contributions in particular, have been shown to increase project success and viability¹⁹” (Lines 179-184).*

1.18 Comment: The last part of the sentence: “as greater in situ contributions have been shown to increase project success and viability” also needs clarifying. Presumably project success is increased through greater financial contributions but that is not what has been measured in this study.

1.18 Response: *This has been changed to clarify the issues raised, it now reads: “The flagship approach potentially results in increased global wildlife conservation, as greater financial in situ contributions in particular, have been shown to increase project success and viability¹⁹”. (Lines 182-184).*

1.19 Comment: L180-181: Correct reference? Possibly refers to Hutchins et al. 1995. Suggest change to “Historically, personal preferences, availability, and competition between institutions were the main determinants of collection composition”.

1.19 Response: *This reference has been changed to Hutchins et al., 1995. Text also changed to “Historically, personal preferences, availability, and competition between institutions were the main determinants of collection composition¹³” (Lines 216-218) as suggested.*

1.20 Comment: L182: Suggest “decisions are also shaped by...” or “largely shaped”.

1.20 Response: *Changed to “composition decisions are largely shaped by...” (Line 218) as suggested.*

1.21 Comment: L190: Reference 13?

1.21 Response: *This has been changed from reference 14 (Bowkett, 2014) to references 12 (Whitworth, 2012) and 13 (Hutchins et al., 1995).*

1.22 Comment: L218: Would be interesting to know how safari parks and similar drive-through animal parks were classified.

1.22 Response: *Due to the fact that ‘safari parks’ and similar drive-through animal parks are relatively few in number and often contain additional walk-through components, similar to traditional zoos, they were treated the same as other institutions. This is now mentioned in the methods as follows: “Safari parks and similar drive-through animal parks were treated the same as other institutions” (Lines 262 – 263).*

Reviewer #2 (Remarks to the Author):

2.1 Comment: This is a well written paper that describes a significant contribution to an issue that really demands a critical analysis. Such a critical analysis is challenging because the issue of maintaining animals in zoos and the potential of zoos to contribute to conservation, whether in situ or ex situ, or surrounded by very strong

views, both for and against. At the same time, however, there is a real need for marshalling resources for in situ conservation, as is clear from both the recent IPBES Global Assessment and will be clear for species particularly in the reporting for the present Convention on Biological Diversity Strategic Plan, which concludes in 2020.

2.1 Response: *We thank reviewer #2 for their supportive comments.*

2.2 Comment: My comments are rather minor, but would, I think help with increasing confidence in the framing and design on the study. First, although there is much written about various aspects of zoos and captive management, little is in what might be considered front line journals. There is therefore quite a diversity of sources cited in the article and they are presented as sources of equivalent quality and merit. For example, an approach that proposed in the WAZA Magazine is cited, whereas the IUCN ex situ management guidelines (with a supporting publication in Conservation Letters) is not. I am not entirely sure of the veracity of some of the references to zoo species journals and would suggest that the authors consider the extent to which they really do have confidence in them.

2.2 Response: *Although a lot of interest has been expressed in this subject area, little actual research has been conducted and therefore we found a sparsity of knowledge surrounding the topic. This is compounded by the contemporary social aspect of this topic, which is often not considered of high scientific importance. As a result, we relied on a variety of sources when developing the model and expanding themes. In the example cited, we agree that the WAZA magazine is not an ideal reference, however this is the only thorough description of the IUCN SSC CPSG One Plan Approach, to our knowledge. We have modified this to also include the IUCN SSC Guidelines on the Use of Ex situ Management for Species Conservation. (L166-168)*

Although the sources cited are variable we thoroughly test the putative causal pathways and theoretical framework generated from these sources and we believe that this work has important implications. We believe that ignoring these sources of information would lead to a depauperate understanding of the full network of pathways affecting attendance and in situ contributions and hence potentially lead to missing important pathways.

2.3 Comment: Second, there is the option, although I accept word limits, to briefly emphasise the policy context. The EU Zoos Directive has been through a REFIT evaluation, IUCN has relatively new guidelines, and CBD has a clear focus on species conservation (Target 12) that was deemed to be 'moving away from the target' at the mid-term review of the present Strategic Plan in 2014. All of which pleads for the analysis presented here.

2.3 Response: *We thank Reviewer #2 for seeing the policy relevance of this topic. We tried to include a policy-relevant element by including the World Zoo and Aquarium Conservation Strategy and the IUCN SSC CPSG One Plan Approach. We have changed this to also include the Aichi Biodiversity Targets as follows: "This reflects the 'flagship' and the "One Plan" conservation approaches, both of which ultimately contribute to Target 12 (conservation of species) of the United Nations Convention on Biological Diversity Aichi Biodiversity Targets²²" (Lines 214-216).*

2.4 Comment: Finally, it would have been helpful to see a more discursive approach to defining the model. This is too easily dispatched in lines 204-205 and so does not

inspire confidence that the literature review has led to a defensible model. The pathways are stated but not justified and so there is a need to have a process posited for each pathway. Without this it could be argued that it just becomes an exercise in finding the best SEM across a big square correlation matrix. For example:

we assumed that visits to zoos would be positively associated with numbers of visits to zoos because big animals are more charismatic and are likely to attract more visitors (as an example). Then, for example, Smith (1990) demonstrated that size matters.

This would help show that the model is not based on a series of one off relationships but the study of the whole system. Explicitly justifying each of the single variable relationships helps make clear the justification for the creation of the model. The effect on the non-SEM specialist reader, like me, would be to show that the model pathway is based on an explicitly defined underlying process.

2.4 Response: *Given the complexity of the system in question, there has yet to be a systematic evaluation of the whole system for us to draw on, and this is what we try to correct in this manuscript. However singular relationships between the variables used were found as part of our literature review, suggesting several clear underlying processes. These pre-defined relationships were collectively used in the development of the a priori meta-model, in conjunction with our proposed hypotheses. These evidence-based relationships are cited and can be seen in Supplementary Figure 1. Given the number and complexity of all the studies cited in this figure, we have elaborated and explained in detail the origin of the pathways depicted in the a priori meta-model in the Supplementary Methods. To clarify this, we have reworded this section in the main manuscript as follows: “We conducted a literature review of the relationships between institutional attendance, zoo species composition and in situ contributions. Based on this prior theoretical knowledge and proposed causal relationships we developed a hypothetical a priori meta-model^{24, 25} (Supplementary Figure 1). A thorough description of both the prior theoretical knowledge and proposed causal relationships used to generate the a priori meta-model depicted in Supplementary Figure 1 are explained in the Supplementary Methods.” (Lines 239-245). We hope this clarifies any issues regarding what the meta-model is and how it was developed.*

Reviewer #3 (Remarks to the Author):

3.01 Comment: This is an interesting and generally well-written paper that utilizes a large data set to assess the relationships between zoo ‘size’ (= area and number of visitors) and various variables linked to the role of zoos in conservation. The relationships are explored using structured equation modelling, which I am not familiar with, but the links between the variables are nicely displayed I enjoyed reading the paper and believe that there is information presented here that will be of some interest to the zoo community and those involved with ex-situ conservation.

3.01 Response: *We thank Reviewer #3 for their supportive comments.*

3.02 Comment: However, despite the sophisticated analysis I wonder if there is anything that is really new here? My guess is that the main finding that bigger zoos are more involved with in-situ conservation will come as no surprise to the zoo

community. Indeed, I suspect that similar relationships would emerge with almost any business, i.e. bigger businesses are likely to invest more in charitable and corporate social responsibility activities.

3.02 Response: *We would like to clarify that bigger zoos, only in terms of institutional attendance and not actual institutional area, contribute more to in situ conservation. Although this finding may sound intuitive and unsurprising, to our knowledge, this is the first time anyone has been able to support this assumption with quantitative analyses at any scale. In general, there is some prior evidence in the literature that supports several of the bi-variate links which are part of our SEM (i.e. see the relationships cited in Supplementary Figure 1, the a priori meta-model); however, the direct and indirect links between the parts of the model have never been tested previously. In this way we take fragments of evidence scattered in the literature to build a relatively complex model of a range of potential drivers of attendance and in situ conservation contributions and test this using a large and robust data set. While some of our results confirm fragments of prior knowledge, the complexity of direct and indirect effects of drivers of attendance and in situ conservation contributions, were not known prior to our research. Understanding what is driving visitor attendance, and therefore in situ contributions, is an important step if we are going to increase the in situ contributions of zoos globally.*

3.03 Comment: A secondary result that apparently emerges is that an alternative strategy would be for zoos to focus on more, unique smaller species, but I can't see where this result emerges from the results that are presented. These points are elaborated on in some of the specific comments below.

3.03 Response: *As shown in Figure 2, there are direct positive relationships between collection uniqueness (i.e. Dissimilarity), total number of animals and visitor attendance. This suggests that having lots of unique, smaller animals is an alternative way to increase visitor attendance that does not rely on increasing species body mass. In order to clearly communicate our results, we have calculated the net total effect of each variable on attendance (see Figure 1, Table 2 and Supplementary Table 4) which enables a clearer comparison between variables, which includes both their direct and indirect effects.*

Specific queries:

3.04 Comment: Lines: 34-36: This is the key result. But is it really surprising to find that zoos with many large animals get more visitors and contribute to more in-situ conservation projects?

3.04 Response: *Although this result may sound intuitive and unsurprising, to our knowledge, this is the first time anyone has been able to support this assumption with data on any scale. In addition, the effects of number of animals and mean species body mass are only two direct correlates of attendance among many. Here we also show that many other direct and indirect correlates of attendance may also be effective in driving attendance and in situ contributions.*

3.05 Comment: Line 50: It is rather over-simplistic to suggest that in-situ conservation activities are funded through paying visitors. There are various models by which such activities are funded, e.g. external grants, conservation campaigns, special events, zoo memberships, entrepreneurial activity etc.

3.05 Response: *Although zoos pursue many avenues to generate funding for their in situ conservation activities, their primary source of income is often through paying*

visitors. Therefore, in the absence of a way to quantify these alternative income-generating strategies on a global scale, we use visitor attendance as a proxy of income that could fund in situ activities. This is now acknowledged in the text as follows: “In the absence of available revenue data, we use visitor attendance as a proxy of income to potentially fund in situ activities” (Lines 253 – 254). We also acknowledge this point in the abstract: “Zoos contribute substantial resources to in situ conservation projects in natural habitats using revenue from visitor attendance, as well as other sources.” (L20-21)

3.06 Comment: Lines 52-54: Needs slight rewording, as it suggests that ‘Strategic Collection Planning’ is about large vertebrates.

3.06 Response: We agree that Strategic Collection Planning may have several meanings depending on the author and context. ‘Strategic Collection Planning’ as outlined by Hutchins et al. (1995) states that “zoos should focus their long-term breeding programs primarily on flagship species”. Although seemingly non-charismatic species can become flagship species e.g. fish, amphibians etc., according to Hutchins et al. (1995), states that a focus on “larger, charismatic vertebrates... [is] a legitimate and potentially highly effective conservation strategy”. Therefore, we deduce that ‘Strategic Collection Planning’ focuses on flagship species in this context. To avoid confusion we have removed the wording “Strategic Collection Planning” from the manuscript and replaced it with “the flagship approach” and a corresponding reference to Hutchins et al. (1995).

3.07 Comment: Lines 58-59: I’m not entirely sure what the comment about ‘bivariate relationships’ is referring to. Presumably, it is trying to suggest that multiple variables need to be considered together when exploring these relationships? Perhaps tighten up.

3.07 Response: Yes, we are suggesting that multiple variables must be considered simultaneously when exploring these relationships. This has been changed to “While the direct effects of various factors on attendance have been the focus of previous studies, such approach’s fail to capture the complexity of potential indirect drivers of, and trade-offs for, visitor attendance. A framework linking the direct and indirect effects of collection composition variables on conservation outcomes, such as in situ contributions, would allow for more informed collection planning decisions and policy formation.” (Lines 60-65) to help clarify.

3.08 Comment: Line 73: A slight rewording could tighten this up, i.e. ‘We found that zoos with high attendance contribute to more in situ conservation projects’.

3.08 Response: Changed to “We found that zoos with high attendance contribute to more in situ conservation projects” (Line 82-83) as suggested.

3.09 Comment: Lines 76-78: Although tangential to the main point of the paper, a qualifier here is that socio-economic variables may play a lesser role in influencing attendance if admission to the zoo is free or highly subsidised. This may be the case in countries where zoos are run by national or local government.

3.09 Response: We agree that entry fee and fee subsidisation could play an important role in determining visitor attendance and we would have liked to include such information in this manuscript. However, this information is not available for

many institutions. In addition, the known number of 'free' institutions is extremely low in number, with only nine institutions estimated to be free in the USA.

3.10 Comment: Lines 88-89: I can't follow this sentence as the first part appears to contradict the second part.

3.10 Response: *We agree that these sentences are confusing. This has been reworded as follows: "Mammal species richness alone had a direct positive effect on attendance as well as multiple indirect positive effects through the total number of animals. Mammal species richness also had a small negative effect on attendance mediated by species richness. However, the total effect of mammal species richness on attendance was greater than that of overall species richness (Figure 1 and Table 2), suggesting mammals are more important in driving visitor attendance than other taxonomic groups" (Lines 127-132).*

3.11 Comment: Lines 97-100: I think a little more caution is needed here about cause and effect. The analyses show relationships, but the old adage that correlation does not equal causation needs to be considered. There may be some subtle nuances within the data that are playing roles here.

3.11 Response: *This has been changed to "We conclude that the absence of large vertebrates from collections may not necessarily result in reduced in situ project activity, presuming institutional attendance can be maintained in their absence through an increase in collection dissimilarity, species richness and/or total number of animals" (Lines 155-159) to temper causality.*

3.12 Comment: Lines 104-105: Again, I find the result that greater investment in ex situ conservation is positively related to in situ conservation not very surprising. Indeed, this is the direction that zoos have been heading in for some years, and is enshrined within the World Zoo Conservation Strategy. There are many good existing case studies of this within the literature.

3.12 Response: *We agree that this result is not surprising and that many good case studies exist to show this. However, this result is the first to confirm that these are not simply isolated incidences and actually demonstrate these relationships are occurring at a global scale.*

3.13 Comment: Figure 1 (and generally for the methods): I am not familiar with the SEM framework for analysing such relationships, but I did wonder what the advantages might be over a traditional GLMM?

3.13 Response: *The primary benefit of using SEM over a traditional GLMM is the ability to investigate multiple direct and indirect relationships within a system simultaneously. GLMM would have enabled us to test the effects of multiple explanatory variables on attendance but would not have enabled us to test the indirect effects of one explanatory variable driving another explanatory variable. For example, SEM enables us to assess the effects of body size on attendance directly but also through its correlation with total number of animals (which is negative). This aspect of SEM is highlighted in the text as follows: "Structural Equation Modelling (SEM) integrates multivariate relationships, testing both direct and indirect effects within a system²³" (Lines 237-238).*

3.14 Comment: Table 1: 'Number of in situ projects' is used as the variable to measure zoo involvement in field conservation. Although some relationships with this

variable apparently emerge, this is a fairly coarse measure of in situ activity. Zoos have different models for investing in in situ conservation: some support a wide range of projects with modest funds, while others invest more substantially in a smaller number of projects. So my points are that (1) some relationships may be masked by using 'number of projects'; and (2) conservation spend on in situ projects may actually reveal other relationships.

3.14 Response: *This point is also raised by Reviewer 1, see Responses 1.02 and 1.03.*

3.15 Comment: Lines 146-147: Implies that Strategic Collection Planning is all about flagship species – needs rewording to take account of the fact that it is just one element.

3.15 Response: *See Response 3.06.*

3.16 Comment: Lines 156-157: The alternative strategy of exhibiting numerous, unique, smaller-bodied species emerges here as another key finding, but I cannot see where this is explicitly produced from the results that are presented earlier.

3.16 Response: *As mentioned in response 3.03 the pathways in our SEM, shown in Figure 2, demonstrate a negative relationship between mean body size and species richness, dissimilarity and total animal numbers. Hence one route to increased attendance is to increase the mean body size of animals. An alternative way to increase visitor attendance that does not rely on increasing species body mass is to have lots of unique, smaller animals.*

3.17 Comment: Lines 165-166: Again, I cannot see where the results show this conflict. This seems to be a speculative inference of the results perhaps, rather than a result in itself.

3.17 Response: *That is correct, the results do not directly show this conflict and it is an inference of the results in the context of known population management recommendations and observed global welfare trends. For example, the results show that collection dissimilarity is positively correlated with institutional attendance. However, here we highlight that population management recommendations often encourage institutions to consolidate their collections in order to enhance management efficacy, which would ultimately lead to the uniformity of collections, which in turn would decrease collection dissimilarity and therefore decrease institutional attendance (Conde et al., 2011).*

3.18 Comment: Lines 166-170: This implies that there is a trade-off between collections avoiding similarity to avoid competition for paying visitors, while breeding programmes encourage collections to consolidate collections on similar species as this makes species management more efficient. This is probably true to some extent, but again there are subtle nuances at play here. Many breeding programmes are managed on a European-wide scale (e.g. via EAZA) and it is unlikely that, for example, zoos in the UK and Germany would compete with each other because they hold the same species. Likewise, on a regional scale, there is already clear dissimilarity on regional scales. For example, East Anglia is an area with a historically high density of zoos, and back in the 1980s there were major collections within about 50 miles of each other focusing on Europe, Latin America and Asia. Although some of these collections no longer exist the specialisms within the regions

persist. So in sum, I think this trade-off is not really a big issue when it comes to collection planning because there are more important drivers at play here.

3.18 Response: *We agree that collection dissimilarity is only one part of the complex system of attendance determinants and that there are more important drivers at play. This is highlighted by the fact that collection dissimilarity has one of the lowest effect sizes on visitor attendance of all variables. However, it is still important to consider the implications of collection dissimilarity on collection planning and visitor attendance.*

3.19 Comment: Lines 184-186: Overall, I think this is rather an over-statement of the relevance of the results to policy-makers, collection planners and conservation. The results will certainly be of interest, but the bigger zoos have marketing teams that have a strong handle on what drives visitor attendance and conservation teams that appraise what are the most cost-effective in-situ projects to get involved with given zoo income streams and strategic objectives.

3.19 Response: *Although on the surface this issue may seem zoo-specific, zoos and aquariums are under increasing social and political pressure. This is evinced by recent legal battles in the USA, Canada and Europe surrounding the keeping of large, charismatic vertebrates in captivity, particularly cetaceans. This issue has also been raised as part of the recent review of the EU Zoos Directive, a document which highlights the contribution of zoos to the United Nations Convention on Biological Diversity Aichi Biodiversity Targets and to the 2020 EU Biodiversity Strategy. Therefore, we consider this global analysis and the results presented here to have important relevance for policy-makers, collection planners and conservationists.*

3.20 Comment: Lines 190-191: As mentioned above, I think this is already being done on a regional scale. Witness the differences between collections in areas such as East Anglia and Kent.

3.20 Response: *Indeed, our analysis reveals pre-existing relationships. The suggestion that “the exhibition of large numbers of animals in collections that are dissimilar to other zoos is a viable alternative strategy” (Lines 226-227) likely stems from the existence of the regional differences mentioned.*

3.21 Comment: Lines 218-219: Removal of aquariums from the data set is perfectly justified, but what about other specialist zoos, such as bird parks and those with a geographical specialism?

3.21 Response: *In this study we assessed the mammalian, avian, reptilian and amphibian species held within collections Specialist zoos were retained in the analyses as they represent unique institutions with the potential to increase visitor attendance through their unique collections. We have accounted for these institutions and their unique collections in our analyses by specifically investigating collection taxonomic diversity, collection dissimilarity (relative to other institutions) and threatened species representation as explanatory variables for institutional attendance.*

Reviewer #4 (Remarks to the Author):

I was asked to evaluate the methodology of this paper. As I am by no means an expert on zoos or conservation planning, I will not comment on content unrelated to the methodology. That being said, I have several concerns regarding the way the Structural Equation Models were performed. See my comments below.

Reviewer 4 general Response: *We appreciate the thorough statistical review provided and have responded to individual comments below.*

4.01 Comment: Lines 76-82: “Collection composition variables (no. of animals, species body mass, compositional dissimilarity and mammal species richness) are more important in determining attendance than socio-economic variables.”

I am not so sure about this. Given the many indirect pathways linking attendance to collection composition and socio-economic variables, it would be quite helpful to use the standardized path coefficients to calculate total effects. In this way, you could for example calculate the total effect of mean species body mass on attendance, which is here the sum of:

- the direct effect on attendance
- the indirect effect mediated by dissimilarity
- the indirect effect via species richness and dissimilarity
- the indirect effect via species richness and total animals
- the indirect effect via species richness, total animals and dissimilarity

Of course, this ignores pathways via correlated exogenous variables, so I would also include covariances among exogenous variables, unless these were fixed, which is not clear to me (see also next comment), and which I would not recommend unless there are strong theoretical reasons to assume these are truly uncorrelated.

The point is: it may well be that the total effect of collection composition variables on attendance is weaker than that of socio-economic variables.

4.01 Response: *We agree, these indirect relationships and their associated values are key to understanding the system in question. To clarify the effect sizes, we have taken up the reviewer’s suggestion. We have calculated the total effect for each relationship, summing all the direct and all indirect effects and included them in our manuscript as part of Table 2 and Supplementary Table 4. We now display this information graphically in Figure 1, with a simplified version of our SEM, depicting only the total effect sizes. From these values it is clear that the total effect of collection composition variables on attendance remains much stronger than that of socio-economic variables. We include the calculation of Total Effects in the new Supplementary Code Rmarkdown file to enable these to be reproduced.*

In addition, covariances among exogenous variables were not fixed. Residual covariances are now shown in Supplementary Tables 5 and 6.

4.02 Comment: “The total number of animals had the largest direct positive effect on attendance, followed by abundance weighted mean species body mass, with compositional dissimilarity and mammal species richness having smaller direct positive effects (Fig. 1). Ultimately, there is strong evidence that the presence of larger-bodied species is strongly related to increased institutional attendance and in turn, in situ contributions.” I have two issues with this. First, the second-largest direct positive effect on attendance is that of “10 km population”, not that of “mean species body mass”.

4.02 Response: *The statement was meant to imply that of the collection composition variables assessed (i.e. not focusing on the socio-economic variables), mean species body mass had the second greatest direct positive effect on attendance. This sentence has been reworded to “Of the collection composition variables, the total number of animals had the largest direct positive effect on attendance, followed by abundance weighted mean species body mass, with compositional dissimilarity and mammal species richness having smaller direct positive effects (Figure 2) (Lines 116-119) to clarify this point.*

4.03 Comment: Second, mean species body mass has strong negative effects on dissimilarity and species richness, both of which increase attendance. So again, without quantifying these indirect effects, I am not sure whether mean body mass is related to increased institutional attendance.

4.03 Response: *Although mean species body mass has strong negative effects on other variables, the total effect of mean species body mass on attendance (summing both direct and indirect effects) remains positive. This suggests that attendance is positively correlated with mean species body mass for an institution. Total effect values have also been added to Table 2 and Supplementary Table 4 for all relationships and are shown graphically in Figure 1.*

4.04 Comment: Line 89-93: “Mammal species richness had a direct positive effect on attendance as well as an indirect positive effect through the total number of animals. Mammal species richness could have a small negative effect on attendance if the increase in total species richness does not lead to increased total number of animals.”

1) The indirect positive effect is not only through the total number of animals, but also through the total species richness, which positively influences total number of animals. Moreover, there could be an indirect negative effect of mammal species richness on attendance by increasing total species richness, which in turn has a direct negative relationship with attendance.

4.04 Response: *We agree that these sentences are confusing, and that Mammal Species Richness has both positive and negative indirect effects on visitor attendance. This has been reworded as follows: “Mammal species richness alone had a direct positive effect on attendance as well as multiple indirect positive effects through the total number of animals. Mammal species richness also had a small negative effect on attendance mediated by species richness. However, the total effect of mammal species richness on attendance was greater than that of overall species richness (Figure 1 and Table 2), suggesting mammals are more important in driving visitor attendance than other taxonomic groups.” (Lines 127-132).*

4.05 Comment: 2) The increase in total species richness does lead to increased total numbers of animals, based on your model, so I do not understand the meaning of the second sentence here.

4.05 Response: *We have now amended the sentence as described in response 4.04*

4.06 Comment: Line 95: “No support was found for linking species body mass directly with in situ project activity”. But this pathway is not present in the model? Or did you only depict significant pathways? I would suggest to display non-significant paths, for example with dashed arrows.

4.06 Response: *This pathway is not presented in the final model as no statistical support was found for the inclusion of it. However, as stated in the text, tests of mediation were performed on mediated pathways to ensure both direct and indirect effects of variables were justified in both models. For this particular relationship we compared final models with complete mediation, partial mediation and no mediation, with the complete mediation model proving superior. Therefore, although we tested for it, we found no support for linking species body mass directly with in situ project activity (i.e. relationship is completely mediated). The visualisation of all possible tested pathways would make this figure very difficult to read. Instead we direct readers to the a priori meta-model which depicts our original model of relationships (from theory and literature) and can be used to visualise a priori pathways that were tested.*

4.07 Comment: Line 100. Or by having a higher mammal or total species richness, presuming that their total effect on attendance is positive (which you could calculate).

4.07 Response: *This is correct, and we have reworded the sentence as follows to avoid confusion: "...an increase in collection dissimilarity, species richness and/or total number of animals" (Lines 158-159).*

4.08 Comment: Line 138: "See Extended Data Tables 1 and 2 for test statistics and fit indices, standardized path coefficients, significance values and proposed interpretations of causal pathways." This is a bit misleading, as it suggested to me that the path coefficients shown are unstandardized, and the standardized ones can be found in supplementary data. However, they match, so all values are standardized. I think it would be nice to provide unstandardized coefficients as well, in case people want to use your SEM for things like predictions.

4.08 Response: *We have added the line "Path coefficients shown are standardized." to each figure legend to help avoid confusion. The standardization of model coefficients allows for the direct comparison of the relative strengths of predictors across the system. This is also necessary for the calculation of indirect and total effect sizes, as the predictors used occurred on vastly different scales. Reporting of unstandardized coefficients would not allow this and given the complexity of the model already, we think that reporting the unstandardized coefficients could add confusion. Additionally, the lavaan package requires the standardization of variances to an approximately similar scale, therefore we cannot run a model on unstandardized data.*

4.09 Comment: It would also be nice to see the actual P-values rather than the message that they all were below 0.05. There is a big difference between a 0.049 and a 0.001.

4.09 Response: *Table 2 and Supplementary Table 4 have been modified to include the actual P-values. Values lower than 0.001 are shown as "< 0.001".*

4.10 Comment: I would also like to see R2 values for all endogenous variables, and residual variances.

4.10 Response: *R2 values have been added to Table 2 and Supplementary Table 4 for all endogenous variables in both the species presence-absence and abundance adjusted models.*

4.11 Comment: Were residual covariances allowed in the model, and if so, what were their values?

4.11 Response: *Residual covariances were allowed in the model. Their values are now shown in Supplementary Tables 5 and 6.*

4.12 Comment: Were means, variances and covariances of exogenous variables free or fixed?

4.12 Response: *These values were not fixed within the models.*

4.13 Comment: Line 157. “no single optimal collection composition exists”. Based on your model, you could calculate which of the two proposed strategies increases in situ conservation investment most, instead of speculating about it.

4.13 Response: *We can look at the total effects associated with each strategy depicted in the model and this will show that any strategy that increases the total number of animals will have a stronger effect on attendance compared to increasing body mass or any other parameter. However, the costs and effort needed to increase total number of animals is not necessarily equivalent to that needed to increase mean species body mass. Without understanding the economic, logistical and regulatory constraints on changing composition it is not possible to determine an “optimal” composition. We are reluctant to quantify a statistical optimum here as this ignores the complexity of the real system and would be misleading for policy makers and zoo managers. The benefit of our analysis is that it demonstrates that several potential pathways are available.*

4.14 Comment: Line 201. What makes you think that the multivariate relationships in SEM are causal? SEM is not different from any other form of data analysis, in that it cannot replace comparison of experimental manipulation with controls, which is the only way to establish causality. SEM based purely on observed, unmanipulated variables therefore can never prove causality.

4.14 Response: *We agree that SEM cannot establish causal relations from associations alone, and in fact it does not aim to do so. The inference that can be drawn from SEM analyses is derived from the interpretation of parameters as evidence consistent with particular causal processes, based on firm scientific knowledge, previously conducted experimental studies or logical arguments. The goal of SEM is to combine qualitative causal assumptions with empirical data, allowing us to generate quantitative causal conclusions and statistical measures of fit. The ‘fitting’ of the data does not prove the causal assumptions; however, it makes them tentatively more plausible, or is evidence consistent with those causal pathways (Bollen and Pearl, 2013). We do not suggest that SEM based purely on unmanipulated observations alone can prove causality, however we cannot deny the possible causal relationship implied from our causal assumptions and the results obtained. In order to prevent confusion, we have removed the word ‘causal’ from this sentence completely, it now reads as follows: “Structural Equation Modelling (SEM) integrates multivariate relationships, testing both direct and indirect effects within a system²³” (Lines 237-238).*

4.15 Comment: Line 204. The explanation about the meta-model is a bit vague.

4.15 Response: *Given the number and complexity of all the studies cited in the a priori meta-model, we have elaborated and explained in detail the origin of the pathways depicted in the a priori meta-model in the Supplementary Methods. In*

addition, this paragraph has been reworded as follows “We conducted a literature review of the relationships between institutional attendance, zoo species composition and in situ contributions. Based on this prior theoretical knowledge and proposed causal relationships we developed a hypothetical a priori meta-model^{24, 25} (Supplementary Figure 1). A thorough description of both the prior theoretical knowledge and proposed causal relationships used to generate the a priori meta-model depicted in Supplementary Figure 1 are explained in the Supplementary Methods.” (Lines 239-245). We hope this clarifies any issues regarding what the meta-model is and how it was developed.

4.16 Comment: Line 258. Analyses. Several aspects of the used analyses are unclear to me. You build an a priori meta-model (whatever that means), represented in Extended Data Figure 1. I guess this is the theoretical model based on prior knowledge.

4.16 Response: *Yes, our “a priori meta-model” is a theoretical model based on prior knowledge, this is similar to the wording used by Grace et al., 2010, who define the meta-model as “models that represent general relationships among multiple theoretical constructs while omitting statistical detail”. See Response 4.15 for rewording of paragraph to avoid confusion and the inclusion of a reference to Grace et al.’s publication.*

4.17 Comment: However, it is very unclear how you went from this model to the final model represented in Figure 1.

4.17 Response: *see response 4.21*

4.18 Comment: The caption of Extended Data Figure 4 states: “grey dotted lines indicate relationships that were not directly assessed, but incorporated into analyses” What does this mean? How could they be incorporated into analyses without being assessed?

4.18 Response: *This means that institutions were included as a nested variable within countries in the model, but we did not assess the effect of countries directly. This is mentioned in the methods section. As a result, this caption has been changed to “grey dotted lines indicate relationships that were not assessed” to avoid confusion. (L720-721)*

4.19 Comment: There are variables in this model that do not occur in the final model and vice versa, or is “collection taxonomy” the same as “dissimilarity”? Or are these latent variables with multiple indicators?

4.19 Response: *The variables in the a priori meta-model that do not appear in the final models (e.g. ‘Collection Taxonomy’) are general concepts used to help define the meta-model. As the individual studies cited in the a priori meta-model measured very specific relationships, we wanted to simplify their measures into broader themes. This procedure is now explained in the Supplementary Methods: “However, these are general concepts used to help define the a priori meta-model and they do not appear in the final model. Rather than create single indicator latent variables, we place the exact variables measured into our final models. So, instead of placing ‘Collection Taxonomy’ in the final model as a single indicator latent variable, we used the exact measured variable i.e. “Mammal Species Richness [per institution]”. In this manner we use specific relationships from the literature to define general concepts in the a priori meta-model, these general concepts are then represented in the final*

models by specific measurements once again. This explains why certain variables in the a priori meta-model do not appear in the final models (e.g. 'Collection Taxonomy')." (L611-621)

4.20 Comment: Supplementary Tables 1 and 2 lists the variables for Attendance and In Situ models, but variables such as "50km_pop" and "Mean.Jaccard.Dissim.P.A." do not occur in any of the models you present in this manuscript?

4.20 Response: *These are variables that were included in the initial models, however no statistical support was found for their retention in the final models. They are included in Supplementary Tables 7 and 8 to show we assessed them, even if they were not retained. The following line has been added to each table caption to avoid confusion: "Variables assessed. but not retained in the final models are also presented."*

4.21 Comment: The only clue the reader gets is that "Modifications to the model with pathways removed or inserted were based on model-data fit. All modifications were made based on logical interpretations of the data resulting in the elimination of certain variables." What was the exact basis for doing so? Modification indices? What were the cutoff levels? In principle, each modification you make to a hypothetical model of which the overall fit fails upon testing, requires new, independent data. Results from a modified model must be considered provisional only. Adding and removing paths until the overall model fit is adequate, or all paths become significant, are clear examples of overfitting.

4.21 Response: *We have updated this paragraph to better describe how we created our original model and chose our final model:*

"We combined prior theoretical knowledge and proposed causal relationships to create the a priori SEM meta-model (Supplementary Figure 1 and Supplementary Methods). The meta-model captured all evidence-based relationships that we found in our literature review and all plausible and suspected predictors of attendance that we hypothesised. This model was then refined to create the final model depicted in Figure 2 using the approach described in Grace et al. ³⁷ and similar to that implemented in Grace et al. ²⁵. In summary, the a priori meta-model was modified through addition and deletion of pathways using model-data fit procedures to produce a range of plausible alternative models which were compared using AICc values. All modifications to the model, with pathways removed or inserted, were based on quantitative recommendations, theoretical intuition and model-data fit. Model-data fit was assessed using a combination of absolute fit indices (e.g. Standardised Root Mean Square Residual) and incremental fit indices (e.g. Comparative Fit Index), to account for the differential sensitivity of fit indices to data distribution, model size and sample size³⁸. Modification indices were used to guide the addition of suspected pathways, with a standard cut-off level for the chi-square test criterion of 3.84³⁹. Highest value modification indices were considered first, however as modification indices do not take into account whether or not relationships make theoretical sense, intuitive theoretical relationships were also considered, as outlined in the Supplementary R Code provided. Following the addition of these pathways, p-values were then used to identify potentially unsupported pathways, with a threshold of 0.05. Highest p-values were considered first for removal. Overall

model selection from the pool of competing models was achieved using AICc values⁴⁰, with a threshold of more than two AICc units lower than the nearest competing model being considered sufficient for model selection. The AICc values of competing models are shown in Supplementary Table 1. The final selected Attendance model was validated, using four random subsets of the existing data (n = 200 each time), to ensure parameter estimates were similar when using different datasets from the same sample²³. Institutions were included within countries in the model.” (Lines 331-356).

We hope this clarifies the model selection process. We would also like to highlight that not all model fit measures proved adequate (i.e. cut-off thresholds were not reached for all fit indices) and that although all relationships in the species abundance model were significant, we did not only retain significant pathways. This can be seen in the species presence-absence model, where two relationships (Supplementary Tables 4) were non-significant, yet they were retained and displayed in the final model and figure (Supplementary Figure 5). As model fit indices were used to produce a limited pool of plausible competing models that were subsequently assessed using AICc we do not believe that overfitting occurred. Overall, we follow the approach described in Grace et al., (2015) and similar to the semi-exploratory modelling approach implemented in Grace et al., (2016). Fan et al., (2016) was also used as a guide to help avoid common mistakes in SEM implementation and reporting, with a particular focus on SEM in ecological studies. We have now also included supplementary documentation outlining each of the steps in developing the a priori meta-model, and the associated R code, used to build each of the final SEMs.

4.22 Comment: I have the impression that all “non-supported” links were just the non-significant ones, and subsequently removed from the a priori model, while this is not warranted.

4.22 Response: *As mentioned in Response 4.06, non-significant links were retained and displayed in the final models when there was statistical justification to do so. These non-supported links were all included in the original model, however throughout the model development and modification process, no statistical support was found for their retention.*

4.23 Comment: Line 293 states that significance values can be found in Extended Data Table 2, but I do not see them there.

4.23 Response: *Table 2 and Supplementary Table 4 have been modified to include the actual P-values. Values lower than 0.001 are shown as “< 0.001”.*

4.24 Comment: The explanation of how the Attendance model was linked to the In Situ model (lines 285 to 289) is also not at all clear to me. In general, I would like to see these things spelled out in more detail.

4.24 Response: *Due to different sample sizes, with in situ contribution data only being available for a subset of 119/458 institutions, we developed two models. One looking just at visitor attendance and all the variables (Attendance model, 458 institutions) and a second which also looked at the in situ contributions in addition to all the pathways which were retained in the Attendance model (In Situ model, subset of 119 institutions). We combined the results of these two models into a single figure*

(Figure 2) and used the results of the Attendance model to guide the development of all the Attendance-linked pathways in the In Situ model, due to its larger sample size and therefore higher explanatory power. As a result, we show the attendance pathways from the Attendance model and combine this with the in situ pathways of the In Situ model in Figure 2 to provide a holistic overview of the system. We delineate the boundary of the two models with a yellow box in Figure 2. The text has been changed as follows to help clarify this issue and avoid confusion:

“Two distinct SEM frameworks were tested, the Attendance model and the In Situ model. The Attendance model tested the relationship between visitor attendance and all the various specified variables for 458 institutions globally. This model did not include any in situ contribution data. The In Situ model tested the relationship between visitor attendance, in situ contributions and all the various specified variables for a subset of 119 institutions in North America for which in situ contribution data was available. The results of the Attendance model were used to guide the development of the Attendance linked pathways in the In Situ model as the larger sample size of the Attendance model had higher power. The results of the Attendance model are combined with the results of the In Situ Model in Figure 2, with a yellow box delineating the boundary of the two models. Only the additional in situ pathways of the In Situ model are reported, as all other relationships were derived from the Attendance model due to its higher statistical power” (Lines 312-323).

4.25 Comment: Especially useful would be to have the R code of all the different steps and analyses available in a supplementary file. Then at least the reader could get a better grasp of the whole procedure.

4.25 Response: *Please see the Supplementary R Code provided, by making our analyses fully reproducible we hope that readers will better understand the process.*

4.26 Comment: Line 268. By centering and standardizing, the path coefficients become useless for prediction. Why not just report both unstandardized and standardized coefficients?

4.26 Response: *see response 4.08.*

4.27 Comment: Line 297. “however models using species presence-absence only were not qualitatively different”. I see some clear differences between Fig. 1 and Extended Data Fig. 5. For example, in the latter there is a strong positive link between total animals and dissimilarity and there is a negative link between species richness and dissimilarity, both of which are absent in Fig. 1, and there are other differences. These are clearly qualitative differences.

4.27 Response: *We agree that this may seem confusing and that there are significant differences between the species abundance and species presence-absence models. However similar results and conclusions can be seen from both models. This has been rephrased as follows to avoid confusion “however models using species presence-absence only were also assessed and provided overall similar results and conclusions, with qualitative differences found in only four links per model” (Lines 371-376).*

REVIEWERS' COMMENTS:

Reviewer #1 (Remarks to the Author):

Congratulations to the authors for their thorough revision of the manuscript. While I'm not able to assess the specific changes related to the SEM analysis (raised by Reviewer 4), it is clear that the authors have gone to great efforts to provide further detail and clarify their approach. I am satisfied with all the responses to my own comments contained within the rebuttal and would recommend publication assuming the issues raised by the other reviewers have also been largely addressed.

Andrew Bowkett

Reviewer #3 (Remarks to the Author):

The authors have carried out a comprehensive and revision of the paper and have responded constructively to the previous referees' comments. Several areas of the research have been clarified and are now more clearly presented. It is also clear that improving on some of the proxy variables used in the modeling is not possible because of availability of data. The authors also acknowledge that some of the findings may seem intuitive and unsurprising, but argue that this is the first time that they have been described in a quantitative way. From an academic viewpoint, the search for general trends and patterns in large data sets is a potentially valuable exercise. From an application viewpoint, I am less convinced that such broad-brush analyses using proxy variables will realistically influence policy and practice within the zoo community when (1) the findings largely confirm what is already known/suspected/happening; and (2) there are many other more fine-grained variables specific to individual catchment areas that drive the promotion and marketing of zoos to potential visitors (e.g. demographics, travel time, local/regional economy etc.).

Specific comments as follows:

Lines 22-23: As raised in the earlier referees' comments, the models are not exploring causality here. Instead of stating: 'to develop a model of how zoo composition and socio-economic factors directly and indirectly influence visitor attendance and in situ project activity' it would be more appropriate to state: 'to develop a model of how zoo composition and socio-economic factors are related to visitor attendance and in situ project activity.'

Line 28: The apparent trade-off here needs to be clearer. When you say 'many small animals' do you actually mean 'having a large number of small animals than a small number of large animals.'? Either way, the recommendation to shift towards small animals seems to be listed here as a key finding,

while Whitworth (2012) – which is cited later on - actually showed that small body size was more popular than large body size among participants.

Lines 39-44: This implies that zoos are being encouraged to have a ‘compositional shift’, comprising a focus away from large-bodied animals to small bodied animals. I am not sure that this rationale is true. Most zoos that have increased their focus on amphibians, invertebrates and fish have also retained a range of large-bodied species. The strategy has been to attract the visitors through the gate using charismatic megafauna (rightly or wrongly!) and then engage them with smaller, more threatened taxa, rather than phasing in one group at the expense of another as implied here. If my interpretation of the argument here is incorrect then I recommend this section is tightened up.

Line 52: As raised in the referees’ comments, unless there are published data to support it, I am not sure that the generalization ‘These in situ conservation activities are primarily funded by paying visitors’ is true. The authors state that they could find only nine zoos in North America that are free (this may be higher in some other parts of the world). However, in addition to this issue, zoos have different models for funding in situ conservation. For many zoos paying visitors fund the running costs of the zoo (this can be discerned by scrutinizing income and expenditure accounts that are in the public domain), and conservation work is funded through campaigns and external funding. I think stronger qualification is needed here about using ‘paying visitors’ as a proxy.

Table 1: As specialist collections focusing on birds, amphibians or reptiles were apparently included, presumably these had a score for ‘Mammal Species Richness’ of zero?

Lines 85-87: This states that 6 compositional variables are more important than 2 socio-economic variables. However, in Fig. 1 the two thickest arrows are ‘total animals’ (=compositional variable) and ‘population density’ (=socio-economic variable). How is this apparent disparity between the text and figure reconciled?

Lines 123-125: Why is this contrary to expectations? I would not have expected most zoo visitors to be motivated to visit a zoo by the number of threatened species they are likely to see. This needs to be supported by a reference.

Lines 155-157: Whitworth (2012) has already shown that body size may be an unimportant factor in terms of zoo animal popularity, so this may need citing here.

Line 196: I think it is a bit disingenuous to suggest that zoos have ‘simplistic collection planning and policy formation’. Although some smaller zoos may lack the resources or staff to do it in a rigorous analytical way, evidence of a collection plan is now usually a requirement for a zoo license. Equally, many bigger zoos do have a very perceptive grasp of their catchment areas, visitor demographics, visitor preferences, spend per head and other key variables that motivate reasons for visiting. This is often monitored continuously, and drives collection planning as well as marketing and promotion.

Lines 201-205: As raised by a previous referee, I think this is an oversimplification and is taken out of context. Zoos can maintain dissimilarity and competitiveness on a local scale while consolidating on maintaining fewer species in breeding programmes at a larger, perhaps even continental scale. I don’t think that the reference that is cited here (Conde et al. 2011) is suggesting that there is a conflict between the two so more careful citation is recommended.

Lines 226-227: As stated in the response to referees, perhaps state that there is evidence that zoos are already doing this in some regions to maintain their competitiveness and coexistence, rather than intimating that this is something emerging from the results that they have not already thought about.

Line 260: I can understand why aquariums were excluded to prevent bias. But then bird and reptile/amphibian collections were apparently retained. Why would aquaria cause bias while other taxon-focused collections not cause bias? Indeed, many aquaria also hold amphibians and reptiles. I think the rationale for inclusion/exclusion needs to be clearer, as does inclusion of 'number of mammal species' as a predictor in the model.

Line 291: Was there a rationale for using a 10 km catchment area for visitors? This will vary considerably between institutions and would different results have been obtained using a different distance?

Reviewer #4 (Remarks to the Author):

I have now carefully considered the authors' responses to my previous comments. They have done a wonderful job clarifying and better describing their methodology. I only have two rather minor comments left at this point:

- Response 4.04. Unless I am missing something, there is only one indirect effect of mammal diversity on visitor attendance through total number of animals. I therefore suggest removing the word "multiple" from this sentence.

- Response 4.13. I still find the use of the word "optimal" slightly misleading, as mathematically it is perfectly possible to determine the most optimal of several good alternative collection compositions, i.e. the one maximizing in situ contributions. I suggest rewording this sentence along these lines: "... that several alternative collection compositions can result in high attendance and in situ contributions..."

Reviewer #1 (Remarks to the Author):

1.01 Comment: Congratulations to the authors for their thorough revision of the manuscript. While I'm not able to assess the specific changes related to the SEM analysis (raised by Reviewer 4), it is clear that the authors have gone to great efforts to provide further detail and clarify their approach. I am satisfied with all the responses to my own comments contained within the rebuttal and would recommend publication assuming the issues raised by the other reviewers have also been largely addressed.

1.01 Response: We once again thank reviewer #1 (Dr. Andrew Bowkett) for his encouraging statement.

Reviewer #3 (Remarks to the Author):

3.01 Comment: The authors have carried out a comprehensive and revision of the paper and have responded constructively to the previous referees' comments. Several areas of the research have been clarified and are now more clearly presented. It is also clear that improving on some of the proxy variables used in the modeling is not possible because of availability of data. The authors also acknowledge that some of the findings may seem intuitive and unsurprising, but argue that this is the first time that they have been described in a quantitative way. From an academic viewpoint, the search for general trends and patterns in large data sets is a potentially valuable exercise.

3.01 Response: We thank Reviewer #3 for their supportive comments.

3.02 Comment: From an application viewpoint, I am less convinced that such broad-brush analyses using proxy variables will realistically influence policy and practice within the zoo community when (1) the findings largely confirm what is already known/suspected/happening; and (2) there are many other more fine-grained variables specific to individual catchment areas that drive the promotion and marketing of zoos to potential visitors (e.g. demographics, travel time, local/regional economy etc.).

3.02 Response: We appreciate the clear limitations of the analysis presented here and agree that further, finer-scaled analyses could potentially reveal greater detail which is currently not in the scope of this analysis.

3.03 Comment: Lines 22-23: As raised in the earlier referees' comments, the models are not exploring causality here. Instead of stating: 'to develop a model of how zoo composition and socio-economic factors directly and indirectly influence visitor attendance and in situ project activity' it would be more appropriate to state: 'to develop a model of how zoo composition and socio-economic factors are related to visitor attendance and in situ project activity.'

3.03 Response: All suggestions of causality have previously been removed from the manuscript as requested. We believe that the phrase 'influence' is appropriate and we are inferring influence from an underlying correlation.

3.04 Comment: Line 28: The apparent trade-off here needs to be clearer. When you say 'many small animals' do you actually mean 'having a large number of small animals than a small number of large animals.'? Either way, the recommendation to shift towards small animals seems to be listed here as a key finding, while Whitworth (2012) – which is cited later on - actually showed that small body size was more popular than large body size among participants.

3.04 Response: The trade-off between Mean Species Body Mass and Total Animals is clear in Figure 2. Both Mean Species Body Mass and Total Animals have positive direct effects on Attendance (Lines 23-25), yet Mean Species Body Mass has a direct negative effect on Total Animals (Lines 26-27). Yes, by saying 'many small animals' we mean having a greater number of smaller animals compared to a smaller number of larger animals. This paper tests the hypothesis that large, charismatic vertebrates are necessary in order to attract visitors, a phenomenon widely believed to be true. We found this to be true, but also found that having many small animals, may also be effective, which supports the findings of Whitworth (2012). Whitworth (2012) is cited throughout this manuscript and in fact, influences our a priori meta-model. However the scope of Whitworth (2012) is limited, compared to the global analysis presented here.

3.05 Comment: Lines 39-44: This implies that zoos are being encouraged to have a 'compositional shift', comprising a focus away from large-bodied animals to small bodied animals. I am not sure that this rationale is true. Most zoos that have increased their focus on amphibians, invertebrates and fish have also retained a range of large-bodied species. The strategy has been to attract the visitors through the gate using charismatic megafauna (rightly or wrongly!) and then engage them with smaller, more threatened taxa, rather than phasing in one group at the expense of another as implied here. If my interpretation of the argument here is incorrect then I recommend this section is tightened up.

3.05 Response: The compositional shift mentioned is a persistent feature in the literature and is supported by references 8 and 9. We do not contest the alternative strategy mentioned (increasing the focus on amphibians, invertebrates and fish while also retaining large-bodied species). In fact, our results support this strategy (Lines 207-209).

3.06 Comment: Line 52: As raised in the referees' comments, unless there are published data to support it, I am not sure that the generalization 'These in situ conservation activities are primarily funded by paying visitors' is true. The authors state that they could find only nine zoos in North America that are free (this may be higher in some other parts of the world). However, in addition to this issue, zoos have different models for funding in situ conservation. For many zoos paying visitors fund the running costs of the zoo (this can be discerned by scrutinizing income and expenditure accounts that are in the public domain), and conservation work is funded through campaigns and external funding. I think stronger qualification is needed here about using 'paying visitors' as a proxy.

3.06 Response: In the absence of any globally standardised data pertaining to the generation of financial revenue for in situ conservation activities, we use the only available globally standardised

proxy, visitor attendance. We have further emphasised this by stating “These in situ conservation activities are primarily funded by paying visitors, in conjunction with other sources” (Lines 52-53).

3.07 Comment: Table 1: As specialist collections focusing on birds, amphibians or reptiles were apparently included, presumably these had a score for ‘Mammal Species Richness’ of zero?

3.07 Response: Yes, this is correct. As can be seen in the Supplementary Data 1, there are five institutions with a corresponding ‘Mammal Species Richness’ value of zero.

3.08 Comment: Lines 85-87: This states that 6 compositional variables are more important than 2 socio-economic variables. However, in Fig. 1 the two thickest arrows are ‘total animals’ (=compositional variable) and ‘population density’ (=socio-economic variable). How is this apparent disparity between the text and figure reconciled?

3.08 Response: This apparent disparity between the text and figure is reconciled in Table 2. Here you can see that the combined ‘Total Effects’ of compositional variables (total no. of animals, total species richness, mammal species richness, compositional dissimilarity and species body mass) on ‘Institution Attendance’ are greater than the combined ‘Total Effects’ of the socio-economic variables (population density and GDP) on ‘Institution Attendance’. Therefore, it is deducible that collection composition variables are more important in determining ‘Institution Attendance’ than socio-economic variables.

3.09 Comment: Lines 123-125: Why is this contrary to expectations? I would not have expected most zoo visitors to be motivated to visit a zoo by the number of threatened species they are likely to see. This needs to be supported by a reference.

3.09 Response: This is contrary to expectations, as Whitworth (2012) has previously shown that ‘rare’ species are more popular than ‘common’ species, and that increased collection popularity (in terms of species) results in increased visitor attendance. This reference (number 12) has been added to Line 121.

3.10 Comment: Lines 155-157: Whitworth (2012) has already shown that body size may be an unimportant factor in terms of zoo animal popularity, so this may need citing here.

3.10 Response: This conclusion and citation are both presented together on lines 207-209.

3.11 Comment: Line 196: I think it is a bit disingenuous to suggest that zoos have ‘simplistic collection planning and policy formation’. Although some smaller zoos may lack the resources or staff to do it in a rigorous analytical way, evidence of a collection plan is now usually a requirement for a zoo license. Equally, many bigger zoos do have a very perceptive grasp of their catchment areas, visitor demographics, visitor preferences, spend per head and other key variables that motivate reasons for visiting. This is often monitored continuously, and drives collection planning as well as marketing and promotion.

3.11 Response: We agree that collection plans are a common, if not ubiquitous, feature of modern institutions, particularly when considering fine-grained metrics as mentioned. However, given the lack of literature available to support the hypotheses tested here, we believe that a collection plan not including the multitude of variables assessed here at a global scale, could be considered simplistic. We have changed the sentence as follows: “Our results indicate the need to consider multiple direct and indirect drivers of attendance to enable the detection of trade-offs and avoid collection planning and policy formation that do not take the full complexity of the system into account” (Lines 178-179).

3.12 Comment: Lines 201-205: As raised by a previous referee, I think this is an oversimplification and is taken out of context. Zoos can maintain dissimilarity and competitiveness on a local scale while consolidating on maintaining fewer species in breeding programmes at a larger, perhaps even continental scale. I don’t think that the reference that is cited here (Conde et al. 2011) is suggesting that there is a conflict between the two so more careful citation is recommended.

3.12 Response: The paper cited (Conde et al., 2011) states that “spatial distribution of ... zoo populations makes management difficult” and suggests that institutions minimise the distance between conspecifics and consolidate their populations in order to improve population sustainability. This sentence has been crafted in conjunction with the lead-author on the paper cited and we therefore believe it is an appropriate representation of their intentions in consideration of our results.

3.13 Comment: Lines 226-227: As stated in the response to referees, perhaps state that there is evidence that zoos are already doing this in some regions to maintain their competitiveness and coexistence, rather than intimating that this is something emerging from the results that they have not already thought about.

3.13 Response: By revealing and stating that “the exhibition of large numbers of animals in collections that are dissimilar to other zoos is a viable alternative strategy” we believe we are already making it clear that institutions are utilising this strategy effectively.

3.14 Comment: Line 260: I can understand why aquariums were excluded to prevent bias. But then bird and reptile/amphibian collections were apparently retained. Why would aquaria cause bias while other taxon-focused collections not cause bias? Indeed, many aquaria also hold amphibians and reptiles. I think the rationale for inclusion/exclusion needs to be clearer, as does inclusion of ‘number of mammal species’ as a predictor in the model.

3.14 Response: Due to the difficulties in recording fish in the Zoological Information Management System (ZIMS), particularly in terms of group management and recording, we were unable to access data at a resolution precise enough to include fish in our analyses. For example, uncertainties over group/individual recording mean we cannot calculate ‘Total Animals’ or ‘Mean Species Body Mass (abundance adjusted)’. We only included institutions for which we had data to accurately represent the vast majority of their living collections, which meant the removal of aquariums, but the retention of bird/ reptile/amphibian collections. The rationale for the inclusion of ‘Mammal Species Richness’ as a predictor in the model is explained under the ‘Proposed Causal Hypotheses’ of Supplementary Note 1. Here it explains that the modelling approach used was semi-exploratory, and that in the

absence of appropriate literary evidence, plausible direct pathways were included. For example, it is repeatedly mentioned in the literature that visitors expect to see large, charismatic mammals, so in the absence of published work to demonstrate this phenomenon, we link 'Mammal Species Richness' to 'Institution Attendance' in our meta-model.

3.15 Comment: Line 291: Was there a rationale for using a 10 km catchment area for visitors? This will vary considerably between institutions and would different results have been obtained using a different distance?

3.15 Response: As can be seen in the Tables 1 and 2 provided in the Description of Additional Supplementary Files, supporting the Supplementary Data, both 50km and 10km catchment areas were included in the analyses, however no statistical support was found for the retention of the 50km catchment area, as outlined in the Supplementary R Code provided.

Reviewer #4 (Remarks to the Author):

4.01 Comment: I have now carefully considered the authors' responses to my previous comments. They have done a wonderful job clarifying and better describing their methodology. I only have two rather minor comments left at this point.

4.01 Response: We sincerely thank reviewer #4 for their kind words and encouragement.

4.02 Comment: Unless I am missing something, there is only one indirect effect of mammal diversity on visitor attendance through total number of animals. I therefore suggest removing the word "multiple" from this sentence.

4.02 Response: As seen in Figure 2, Mammal Species Richness has two indirect effects on Attendance, as shown below:

Mammal Species Richness → Species Richness → Attendance

Mammal Species Richness → Species Richness → Total Animals → Attendance

Therefore, we suggest keeping the current wording.

4.03 Comment: I still find the use of the word "optimal" slightly misleading, as mathematically it is perfectly possible to determine the most optimal of several good alternative collection compositions, i.e. the one maximizing in situ contributions. I suggest rewording this sentence along

these lines: "... that several alternative collection compositions can result in high attendance and in situ contributions..."

4.03 Response: This has been changed to "These alternative correlative pathways influencing attendance and in situ project activity demonstrate that several alternative collection compositions can result in high attendance and in situ contributions, potentially resulting in the future diversification of collection planning strategies" as suggested (Lines 173-176).